# Smoothing Binary Optimization: A Primal-Dual Perspective

## Abstract

Binary optimization is a powerful tool for modeling combinatorial problems, yet scalable and theoretically sound solution methods remain elusive. Conventional solvers often rely on heuristic strategies with weak guarantees or struggle with large-scale instances. In this work, we introduce a novel primal-dual framework that reformulates unconstrained binary optimization as a continuous minimax problem, satisfying a strong max-min property. This reformulation effectively smooths the discrete problem, enabling the application of efficient gradient-based methods. We propose a simultaneous gradient descent-ascent algorithm that is highly parallelizable on GPUs and provably converges to a binary solution in sublinear time. Extensive experiments on large-scale problems—including Max-Cut, MaxSAT, and Maximum Independent Set with up to 50,000 variables—demonstrate that our method identifies high-quality solutions within seconds, significantly outperforming state-of-the-art alternatives.

## 1 Introduction

In this work, we consider the following *binary optimization* problem:

$$\min_{\boldsymbol{x} \in \{0,1\}^n} F(\boldsymbol{x}), \tag{1}$$

where $\boldsymbol{x}$ denotes a binary variable of dimension $n$ and $F : \{0,1\}^n \to \mathbb{R}$ represents an arbitrary real-valued function defined on the binary domain. Problem (1) serves as a general model for a wide range of combinatorial optimization problems. A prominent special case is the *quadratic unconstrained binary optimization* (QUBO), which is inherently equivalent to the Max-Cut problem and capable of characterizing all of the Karp's 21 $\mathcal{NP}$-complete problems (Lucas, 2014). Moreover, *integer linear programs* can be approximately transformed into QUBO by appropriately penalizing the constraints via quadratic terms (Alessandroni et al., 2025). Other classic problems such as *maximum satisfiability* (MaxSAT) (Bacchus et al., 2021) are also encompassed within this framework.

Exact methods for binary optimization typically rely on branch-and-bound frameworks, integrated with various relaxation techniques such as the linear programming-based relaxations (Barahona et al., 1989) and the semi-definite relaxations (Poljak & Rendl, 1995). Recent advances have further enhanced these methods through sophisticated problem reduction techniques and accelerated cutting plane separation (Rehfeldt et al., 2023). However, the inherent $\mathcal{NP}$-hardness fundamentally limits the scalability of exact approaches, which generally remain practical only for instances involving up to a few hundred variables.

Given the limitations of exact methods, research has shifted toward heuristics for large-scale binary optimization. Traditional CPU-based methods involving customized tabu search (Palubeckis, 2006; Wang et al., 2013), simulated annealing (Bertsimas & Tsitsiklis, 1993) and genetic algorithms (Merz & Freisleben, 1999), rely on well-crafted search mechanisms to iteratively explore the solution space. However, their performance is often constrained by sequential execution. Recent hardware progress has enabled significant improvements through GPU-acceleration, and conventional heuristics have benefited. For instance, ABS2 (Nakano et al., 2023) integrates multiple search strategies and genetic operations executed in parallel on GPUs, while FEM (Shen et al., 2025) introduces the simulated annealing method from a continuous view and optimized via GPU implementations. GPU acceleration has also propelled the development of first-order methods, yielding frameworks such as

the sharp-peak penalized continuous relaxation by Zhou et al. (2025) and the sampling-based gradient policy by Chen et al. (2025). Beyond classical computing approaches, quantum annealing on specialized hardware (Berwald, 2019) has emerged as a promising alternative for QUBO. However, the limited number of qubits and noise sensitivity (Zaborniak & de Sousa, 2021) still impede its widespread practical application.

Meanwhile, learning-based methods are gaining substantial traction as a powerful approach to binary optimization. For example, QUBO problems can be naturally represented using graph structures, enabling the application of *graph neural networks* (GNNs) to effectively model variable interactions and achieve strong empirical performance (Schuetz et al., 2022; Ichikawa, 2024). In addition, Tönshoff et al. (2023) introduces a general graph representation for *constraint satisfaction problems* and proposes a GNN-based reinforcement learning approach enhanced with global search actions. Despite their promise, such methods often operate without theoretical guarantees and encounter practical difficulties related to data acquisition and model generalization.

A prominent line of work within this domain *smooths* Problem (1) using a probabilistic framework (Karalias & Loukas, 2020; Qiu et al., 2025): GNNs are leveraged to parameterize an independent distribution over the binary space $\{0, 1\}^n$, and the loss function is subsequently constructed in terms of the expectation of $F$. We note that this approach is essentially to relax Problem (1) to:

$$\min_{\boldsymbol{x} \in [0,1]^n} f(\boldsymbol{x}), \qquad (2)$$

where $f : \mathbb{R}^n \to \mathbb{R}$ is the *multilinear extension* of $F$ (see Section 2 for definition). Although this relaxation preserves theoretical guarantees relative to the original problem (Wang et al., 2022), the non-convex nature of $f$ in Problem (2) still presents significant optimization challenges.

To tackle this challenge, we revisit the binary optimization Problem (1), and reformulate it into a continuous model by characterizing binarity via continuous variables and convex constraint functions. Based on Lagrangian duality, an equivalent *minimax* problem is derived, to which we develop a *gradient descent-ascent* (GDA)-based algorithm for efficient solutions. We remark that by initializing dual variables as positive values, we induce convexity in the primal space, promoting the identification of high-quality solutions. As optimization proceeds, the dual variables gradually decrease, shifting emphasis from convexity promotion to the enforcement of integrality. We refer to this approach as "a Primal-Dual approach for Binary Optimization" (denoted as PDBO), with the overall framework illustrated in Figure 1.

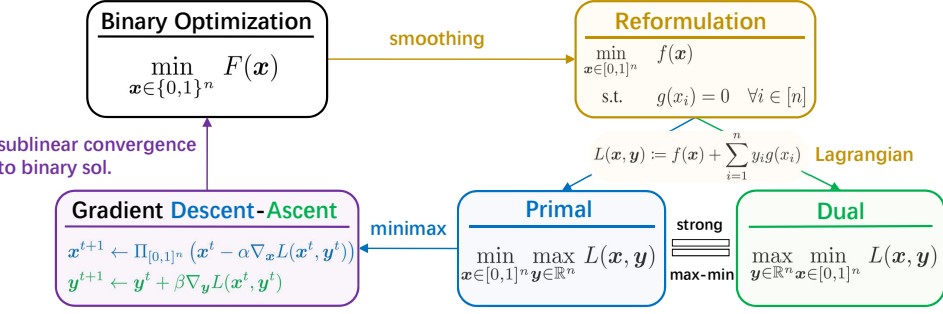

Figure 1: An illustration of PDBO.

The distinct contributions of our work are summarized as follows.

- **Primal-Dual Reformulation.** We introduce a novel framework that reformulates unconstrained binary optimization as a continuous minimax problem, satisfying a strong max-min property. This reformulation smooths the discrete problem, enabling efficient gradient-based optimization.
- **GDA with Guarantees.** We develop a simultaneous GDA-based algorithm that is highly parallelizable on GPUs and converges to binary solutions in sublinear time.
- **Empirical Superiority.** Through extensive experiments on large-scale problems including Max-Cut, MaxSAT, and Maximum Independent Set with up to 50,000 variables, we demonstrate that our method achieves state-of-the-art performance, obtaining high-quality solutions within seconds.

## 2 PRELIMINARIES

This section establishes the core concepts for our framework. We define the multilinear extension, which provides a continuous relaxation for binary optimization, and review its key properties. We also introduce the minimax formalism that underpins our subsequent primal-dual algorithm.

**Definition 1** (Extension). *Let $F : \{0,1\}^n \to \mathbb{R}$ be a function defined on the binary domain. A function $f : [0,1]^n \to \mathbb{R}$ is called an* extension *of $F$ if $f(\boldsymbol{x}) = F(\boldsymbol{x})$ for every $\boldsymbol{x} \in \{0,1\}^n$.*

**Definition 2** (Multilinear Extension). *For any function $F : \{0,1\}^n \to \mathbb{R}$, its multilinear extension $f$ is defined by:*

$$f(\boldsymbol{x}) := \mathbb{E}_{\boldsymbol{\xi} \sim P_{\boldsymbol{x}}}[F(\boldsymbol{\xi})] = \sum_{\boldsymbol{\xi} \in \{0,1\}^n} P_{\boldsymbol{x}}(\boldsymbol{\xi}) \cdot F(\boldsymbol{\xi}),$$

*where for $\boldsymbol{x} \in [0,1]^n$, $P_{\boldsymbol{x}}(\boldsymbol{\xi}) := \prod_{i=1}^{n} x_i^{\xi_i}(1 - x_i)^{1-\xi_i}$ denotes the $n$-dimensional independent Bernoulli distribution with mean vector $\boldsymbol{x}$.*

We remark that $f$ is affine in each variable when all other variables are held fixed. A key property of the multilinear extension is that it preserves the minimum value of the original function:

**Lemma 1.** *Let $f : [0,1]^n \to \mathbb{R}$ be the multilinear extension of $F : \{0,1\}^n \to \mathbb{R}$. Then*

$$\min_{\boldsymbol{x} \in [0,1]^n} f(\boldsymbol{x}) = \min_{\boldsymbol{x} \in \{0,1\}^n} F(\boldsymbol{x}).$$

Lemma 1 establishes the equivalence between Problem (1) and its continuous relaxation (2). However, we note that $f$ is generally non-convex, making the relaxed problem still challenging to solve.

**Proposition 1.** *If $F : \{0,1\}^n \to \mathbb{R}$ is a polynomial function of the form $F(\boldsymbol{x}) = \sum_{\boldsymbol{d}} a_{\boldsymbol{d}} \boldsymbol{x}^{\boldsymbol{d}}$, where $\boldsymbol{x}^{\boldsymbol{d}} := x_1^{d_1} x_2^{d_2} \cdots x_n^{d_n}$, then its multilinear extension is given by:*

$$f(\boldsymbol{x}) = \sum_{\boldsymbol{d}} a_{\boldsymbol{d}} \prod_{i:d_i \neq 0} x_i.$$

Proposition 1 shows that for polynomial objectives (such as QUBO and MaxSAT), the multilinear extension has a compact closed form, avoiding exponential summation. This makes computation tractable for many practical optimization problems.

**Definition 3** (Minimax Problem and Critical Points). *Consider the minimax problem*

$$\min_{\boldsymbol{x} \in \mathcal{X}} \max_{\boldsymbol{y} \in \mathcal{Y}} L(\boldsymbol{x}, \boldsymbol{y}),$$

*where $\mathcal{X} \subseteq \mathbb{R}^n$ and $\mathcal{Y} \subseteq \mathbb{R}^m$ are the domains of $\boldsymbol{x}$ and $\boldsymbol{y}$, respectively. For a point $(\boldsymbol{x}^*, \boldsymbol{y}^*) \in \mathcal{X} \times \mathcal{Y}$:*

- *$(\boldsymbol{x}^*, \boldsymbol{y}^*)$ is a* saddle point *if for all $(\boldsymbol{x}, \boldsymbol{y}) \in \mathcal{X} \times \mathcal{Y}$: $L(\boldsymbol{x}^*, \boldsymbol{y}) \leq L(\boldsymbol{x}^*, \boldsymbol{y}^*) \leq L(\boldsymbol{x}, \boldsymbol{y}^*)$.*

- *$(\boldsymbol{x}^*, \boldsymbol{y}^*)$ is a* stationary point *if $\nabla_{\boldsymbol{x}} L(\boldsymbol{x}^*, \boldsymbol{y}^*) = 0$ and $\nabla_{\boldsymbol{y}} L(\boldsymbol{x}^*, \boldsymbol{y}^*) = 0$.*

**Remark 1.** *A saddle point does not necessarily exist for an arbitrary minimax problem. However, if the strong max-min property holds, i.e.,*

$$\min_{\boldsymbol{x} \in \mathcal{X}} \max_{\boldsymbol{y} \in \mathcal{Y}} L(\boldsymbol{x}, \boldsymbol{y}) = \max_{\boldsymbol{y} \in \mathcal{Y}} \min_{\boldsymbol{x} \in \mathcal{X}} L(\boldsymbol{x}, \boldsymbol{y}) \in \mathbb{R},$$

*then there exists at least one saddle point.*

## 3 METHODOLOGY

In Section 3.1, we introduce a primal-dual framework that reformulates the binary optimization problem (1) as a minimax problem. Then, in Section 3.2, we develop a GDA-based algorithm to solve the resulting minimax formulation and establish its convergence guarantees.

## 3.1 PRIMAL-DUAL REFORMULATION

We begin by reformulating Problem (1) as a constrained continuous optimization problem:

$$
\begin{aligned}
\min_{\boldsymbol{x} \in [0,1]^n} \quad & f(\boldsymbol{x}) \\
\text{s.t.} \quad & g(x_i) = 0 \quad \forall i \in [n],
\end{aligned}
\tag{3}
$$

where $f$ is the multilinear extension of $F$, and $g : [0,1] \to \mathbb{R}$ is a constraint function to enforce binarity such that the condition $g(x_i) = 0$ is equivalent to $x_i \in \{0, 1\}$.

An appropriate function $g$ should satisfy the following sufficient conditions:

- $g$ is convex and continuous on $[0, 1]$, with $g(0) = g(1) = 0$;
- $g$ is differentiable on $(0, 1)$, and $g'(x) = 0$ if and only if $x = \frac{1}{2}$;
- $g$ is symmetric on $[0, 1]$, i.e., $g(x) = g(1 - x)$ for all $x \in [0, 1]$.

These conditions imply that $g(x) < 0$ for all $x \in (0, 1)$ and $g'(x) < 0$ for all $x \in (0, \frac{1}{2})$, and therefore $-g$ can be regarded as a metric of fractionality. Several choices of $g$ have been proposed in the literature. Common examples include an even-degree polynomial $g(x) := (2x - 1)^{2d} - 1$ (Ichikawa, 2024) and the entropy-based function $g(x) := x \log(x) + (1 - x) \log(1 - x)$ (Shen et al., 2025), extended continuously so that $g(0) = g(1) = 0$.

Let $\boldsymbol{y} \in \mathbb{R}^n$ denote the dual variables associated with the constraints in Problem (3). We define the Lagrangian function as:

$$
L(\boldsymbol{x}, \boldsymbol{y}) := f(\boldsymbol{x}) + \sum_{i=1}^{n} y_i g(x_i).
$$

**Proposition 2.** *Let $v^*$ denote the optimal value of Problem (3), then $v^* = \inf_{\boldsymbol{x} \in [0,1]^n} \sup_{\boldsymbol{y} \in \mathbb{R}^n} L(\boldsymbol{x}, \boldsymbol{y})$.*

Proposition 2 characterizes the primal problem as a minimax optimization. The following theorem establishes strong duality for Problem (3).

**Theorem 1.**

$$
\inf_{\boldsymbol{x} \in [0,1]^n} \sup_{\boldsymbol{y} \in \mathbb{R}^n} L(\boldsymbol{x}, \boldsymbol{y}) = \sup_{\boldsymbol{y} \in \mathbb{R}^n} \inf_{\boldsymbol{x} \in [0,1]^n} L(\boldsymbol{x}, \boldsymbol{y}).
\tag{4}
$$

Equation (4) in Theorem 1, known as the *strong max-min property* (Boyd & Vandenberghe, 2004), implies the existence of a *saddle point* of the Lagrangian function that corresponds to an optimal solution of Problem (3). We therefore reformulate Problem (3) as the following equivalent minimax problem:

$$
\min_{\boldsymbol{x} \in [0,1]^n} \max_{\boldsymbol{y} \in \mathbb{R}^n} L(\boldsymbol{x}, \boldsymbol{y}),
\tag{5}
$$

where the order of minimization and maximization can be interchanged due to the strong max-min property.

## 3.2 MINIMAX OPTIMIZATION

The Lagrangian $L(\boldsymbol{x}, \boldsymbol{y})$ is linear in $\boldsymbol{y}$ but not necessarily convex in $\boldsymbol{x}$. Consequently, solving the minimax problem (5) to global optimality remains $\mathcal{NP}$-hard. In this section, we develop a GDA-based algorithm for solving the minimax problem (5) and establish its convergence guarantees.

### 3.2.1 GRADIENT DESCENT-ASCENT

For Problem (5), a classical approach is the GDA method, which alternates between gradient descent on the primal variable $\boldsymbol{x}$ and gradient ascent on the dual variable $\boldsymbol{y}$. The strong max-min property enables *simultaneous* updates as follows:

$$
\begin{aligned}
\boldsymbol{x}^{t+1} &\leftarrow \Pi_{[0,1]^n} \left( \boldsymbol{x}^t - \alpha \cdot \nabla_{\boldsymbol{x}} L(\boldsymbol{x}^t, \boldsymbol{y}^t) \right), \\
\boldsymbol{y}^{t+1} &\leftarrow \boldsymbol{y}^t + \beta \cdot \nabla_{\boldsymbol{y}} L(\boldsymbol{x}^t, \boldsymbol{y}^t),
\end{aligned}
\tag{6}
$$

where $\boldsymbol{x}^t$ and $\boldsymbol{y}^t$ denote the primal and dual variables at iteration $t$, $\Pi_{[0,1]^n}$ denotes the projection onto the hypercube $[0,1]^n$, and $\alpha, \beta > 0$ are the step sizes.

Notably, the projection ensures $\boldsymbol{x}^t \in [0,1]^n$ throughout the optimization, which implies the non-increasing monotonicity of $\{y_i^t\}_t$ since $g(x_i^t) \leq 0$. Moreover, the term $\sum_{i=1}^n (-y_i) \cdot (-g(x_i))$ in $L(\boldsymbol{x}, \boldsymbol{y})$ can be interpreted as an integrality penalty, where the penalty coefficients $-y_i$ increase monotonically during optimization. Moreover, GDA introduces a principled update rule for the penalty coefficients, distinguishing PDBO from traditional penalty methods (Zhou et al., 2025). However, increasing penalties alone does not guarantee integer solutions, as fractional stationary points may persist. For example:

**Example** (Max-Cut). *Let $f(\boldsymbol{x}) \coloneqq \boldsymbol{x}^\top \boldsymbol{W} \boldsymbol{x} - \boldsymbol{1}^\top \boldsymbol{W} \boldsymbol{x}$ and $g(x_i) \coloneqq x_i^2 - x_i$. Then the fractional solution $\boldsymbol{x}^t \coloneqq (\frac{1}{2}, \ldots, \frac{1}{2}) \in \mathbb{R}^n$ yields $\nabla_{\boldsymbol{x}} L(\boldsymbol{x}^t, \boldsymbol{y}^t) = 0$ for any $\boldsymbol{y}^t \in \mathbb{R}^n$.*

To overcome this limitation, we incorporate targeted perturbations to prevent convergence to fractional points. Specifically, given an initial solution $(\boldsymbol{x}^0, \boldsymbol{y}^0) \in [0,1]^n \times \mathbb{R}_{++}^n$, maximum iterations $T_{\max}$, and tolerance $\delta > 0$, we iteratively perform the updates in (6). When $x_i^t$ is closed to and tends to stagnate near $\frac{1}{2}$ (see Step 3), we perturb it to maintain a minimum distance of $\delta$ from $\frac{1}{2}$. The complete procedure is described in Algorithm 1.

---

**Algorithm 1:** PDBO

---

**Input** : Initial solution $(x^0, y^0) \in [0,1]^n \times \mathbb{R}_{++}^n$, stepsizes $(\alpha, \beta) \in \mathbb{R}_{++}^2$, tolerance $\delta > 0$, number of iterations $T_{\max}$

1 **for** $t = 0, 1, 2, \ldots, T_{max} - 1$ **do**
2      **for** $i = 1, 2, ..., n$ **do**
3          **if** $|x_i^t - \frac{1}{2}| \leq \delta$, $|\frac{\partial}{\partial x_i} L(\boldsymbol{x}^t, \boldsymbol{y}^t)| \leq 2\delta$ *and* $y_i^t \leq 0$ **then**
4              $x_i^{t+1} \leftarrow \begin{cases} \frac{1}{2} - \delta & \text{if } x_i^t \leq \frac{1}{2} \\ \frac{1}{2} + \delta & \text{otherwise} \end{cases}$
5          **else**
6              $x_i^{t+1} \leftarrow \Pi_{[0,1]} \left( x_i^t - \alpha \cdot \frac{\partial}{\partial x_i} L(\boldsymbol{x}^t, \boldsymbol{y}^t) \right)$
7          $y_i^{t+1} \leftarrow y_i^t + \beta \cdot g(x_i^t)$

---

We note that for any fixed $\boldsymbol{y}$, the function $f_{\boldsymbol{y}}(\boldsymbol{x}) \coloneqq L(\boldsymbol{x}, \boldsymbol{y})$ remains an *extension* of $F$, with Hessian $\nabla_{\boldsymbol{x}}^2 f_{\boldsymbol{y}}(\boldsymbol{x}) = \nabla_{\boldsymbol{x}}^2 f(\boldsymbol{x}) + \text{diag}(\boldsymbol{y})$. By initializing $\boldsymbol{y}$ as a positive vector with sufficiently large entries, we promote convexity in $\boldsymbol{x}$, encouraging global exploration. During the GDA process, the landscape evolves from convex to non-convex as $\boldsymbol{y}$ decreases. This transition shifts the algorithmic emphasis from solution quality toward integrality enforcement. As established in Section 3.2.2, PDBO guarantees convergence to integer solutions, eliminating the need for rounding.

### 3.2.2 CONVERGENCE ANALYSIS

This section analyzes the convergence of the primal variable $\boldsymbol{x}$ in Algorithm 1 to a binary point, and provides corresponding complexity guarantees. We begin by establishing a lower bound for $y_i^t$.

**Proposition 3.** *Define $\Theta \coloneqq \max_{\boldsymbol{x} \in [0,1]^n} ||\nabla_{\boldsymbol{x}} f(\boldsymbol{x})||_1$ and $b \coloneqq \frac{1+\Theta}{g'(\frac{1}{2} - \delta)} + \left(2 + \lceil \frac{1}{2\alpha} \rceil\right) \cdot \beta \cdot g(\frac{1}{2})$. Then for each $i \in [n]$ and any $t \geq 0$, we have $y_i^t \geq b$.*

Note that for any $i \in [n]$, the sequence $\{y_i^t\}_{t \geq 0}$ is monotone non-increasing, and hence the lower bound leads to the convergence of $\{y_i^t\}_{t \geq 0}$. Furthermore, $\{g(x_i^t)\}_{t \geq 0}$ converges to zero since $\beta g(x_i^t) = y_i^{t+1} - y_i^t$.

**Corollary 1.** *The sequence $\{\boldsymbol{y}^t\}_{t \geq 0}$ converges to some $\boldsymbol{y}^* \in \mathbb{R}^n$ with $y_i^* \geq b$ for any $i \in [n]$.*

**Corollary 2.** *For any $i \in [n]$, the sequence $\{g(x_i^t)\}_{t \geq 0}$ converges to $0$.*

We remark that Corollary 2 implies the convergence of $\boldsymbol{x}$ to a binary point. This holds because $||\nabla_{\boldsymbol{x}} L(\boldsymbol{x}^t, \boldsymbol{y}^t)||$ is bounded, and by choosing a sufficiently small step size $\alpha$, oscillations between binary points can be avoided.

We now establish the iteration complexity for Algorithm 1 to reach a binary solution. Since the primary goal is to solve the binary optimization problem (1), we focus on convergence toward integrality rather than traditional stationarity criteria. The concept is formalized as follows:

**Definition 4.** *Given Problem (3), a point $\boldsymbol{x} \in [0,1]^n$ is called an $\epsilon$-binary point if $-\sum_{i=1}^{n} g(x_i) \leq \epsilon$.*

**Theorem 2.** *The iteration complexity for Algorithm 1 to return an $\epsilon$-binary point is bounded by*

$$\mathcal{O}\left(\frac{\|\boldsymbol{y}^0 - \boldsymbol{y}^*\|_1}{\beta\epsilon}\right)$$

Theorem 2 shows that Algorithm 1 converges to a binary feasible point at a sublinear rate. It should be noted that these results pertain specifically to convergence toward binary points, which differs from classical convergence analyses that are focused on stationary points. More notably, while existing approaches (Qiu et al., 2025; Schuetz et al., 2022) typically depend on carefully designed rounding schemes, PDBO's convergence guarantees eliminate this requirement entirely.

## 4 NUMERICAL EXPERIMENTS

We conduct numerical experiments to evaluate the performance of our proposed PDBO (denoted as `PDBO`) method. We evaluate `PDBO` on four application domains that fit our problem framework, as summarized in Section 4.1. Our method is compared against several state-of-the-art baselines, with results demonstrating significant improvements in both solution quality and computational efficiency. Our code is available at: `https://anonymous.4open.science/r/PD-Bo/`.

### 4.1 APPLICATIONS

We introduce four classic combinatorial optimization problems that align with Problem (1): Max-Cut, Maximum Independent Set (MIS), Max-$k$-SAT, and Max-$k$-Cut.

**Max-Cut.** Given a graph $G := (V, E)$ with adjacency matrix $\mathbf{W}$, the Max-Cut problem seeks to partition the vertex set $V$ into two disjoint subsets that maximize the total weight of edges between them. This can be formulated as the following binary optimization problem:

$$\max_{\boldsymbol{x} \in \{0,1\}^n} \sum_{(i,j) \in E} W_{ij}(x_i + x_j - 2x_i x_j),$$

where $x_i \in \{0,1\}$ indicates the partition assignment of vertex $i$, and $W_{ij}$ denotes the weight of edge $(i,j) \in E$. The expression $x_i + x_j - 2x_i x_j$ equals 1 if $x_i \neq x_j$ (i.e., vertices $i$ and $j$ are in different subsets) and 0 otherwise.

**MIS.** Given a graph $G := (V, E)$, the MIS problem seeks the largest vertex subset $S \subseteq V$ such that no two vertices in $S$ are adjacent. This can be formulated as:

$$\max_{\boldsymbol{x} \in \{0,1\}^n} \quad \sum_{i=1}^{n} x_i$$
$$\text{s.t.} \quad x_i x_j = 0, \quad \forall (i,j) \in E$$

where $x_i \in \{0,1\}$ indicates whether vertex $i$ is included in the independent set. Following the approach of Schuetz et al. (2022); Ichikawa (2024), we reformulate MIS as a QUBO by penalizing the adjacency constraints:

$$\max_{\boldsymbol{x} \in \{0,1\}^n} \sum_{i=1}^{n} x_i - \lambda \cdot \sum_{(i,j) \in E} x_i x_j$$

where $\lambda > 0$ is a penalty parameter. In line with established practice, we set $\lambda = 4$ for all methods to ensure a consistent comparison.

**Max-$k$-SAT.** Given a conjunctive normal form (CNF) formula with $m$ clauses, where each clause contains at most $k$ literals (a variable or its negation), the goal is to find a truth assignment that

maximizes the number of satisfied clauses. This problem can be formulated as the following binary optimization problem:

$$\min_{\boldsymbol{x} \in \{0,1\}^n} \sum_{j=1}^{m} \prod_{l_i \in C_j} p(x_i)$$

where $x_i \in \{0, 1\}$ denotes the truth assignment of the $i$-th variable, $C_j$ denotes the $j$-th clause, and $p(x_i) = x_i$ if literal $l_i$ appears negated in clause $C_j$ and $1 - x_i$ otherwise. The product term $\prod_{l_i \in C_j} p(x_i)$ equals 1 if clause $C_j$ is unsatisfied and 0 if it is satisfied. Thus, the objective function counts the number of unsatisfied clauses. For example, for the CNF formula $(l_1 \vee l_2 \vee \neg l_3) \wedge (\neg l_1 \vee l_3)$, the objective becomes $(1 - x_1)(1 - x_2)x_3 + x_1(1 - x_3)$, which represents the number of unsatisfied clauses.

**Max-$k$-Cut.** The Max-$k$-Cut problem generalizes Max-Cut by partitioning the vertex set $V$ into $k$ disjoint subsets, maximizing the total weight of edges crossing between different subsets. The discrete formulation is:

$$\max_{\boldsymbol{X} \in \mathbb{R}^{k \times n}} \sum_{(i,j) \in E} W_{ij} \left(1 - \boldsymbol{X}_{:,i}^{\top} \boldsymbol{X}_{:,j}\right)$$

$$\text{s.t.} \qquad \boldsymbol{X}_{:,i} \in \{\boldsymbol{e}^{(1)}, \dots, \boldsymbol{e}^{(k)}\}, \qquad \forall i \in [n]$$

where $\boldsymbol{X}_{:,i}$ is a $k$-dimensional one-hot vector indicating the partition assignment of vertex $i$ since $\boldsymbol{e}^{(j)}$ denotes the $j$-th standard basis vector. The expression $1 - \boldsymbol{X}_{:,i}^{\top} \boldsymbol{X}_{:,j}$ equals 1 if vertices $i$ and $j$ belong to different subsets and 0 otherwise. We remark that Max-$k$-Cut can be incorporated into our primal-dual framework with a natural generalization. Please see Appendix B for technical details.

## 4.2 SETUP

**Datasets.** For Max-Cut and Max-$k$-Cut, we use the *Gset* benchmark (Ye, 2003), a widely recognized dataset comprising toroidal, planar, and random graphs with 800 to 20,000 nodes, totaling 71 instances. For MIS, we employ $d$-regular random graphs ($d$-RRGs), where each node has degree $d$, following the experimental setup of Schuetz et al. (2022). To evaluate performance across both sparse and dense graph structures, we consider degrees $d \in \{3, 100\}$ and graph sizes $n \in \{10^4, 5 \times 10^4\}$ nodes. For each $(n, d)$ configuration, we generate 20 independent graph instances to ensure statistical significance. For Max-$k$-SAT, we use the datasets from Tönshoff et al. (2023), which include 3CNF, 4CNF, and 5CNF formulas, each containing 50 instances. This enables consistent and comparable evaluation with existing work.

**Baselines.** We consider two types of baseline algorithms: For traditional heuristics with GPU accelerations, we adopt (i) `ABS2` (Nakano et al., 2023): an evolutionary method featuring three diverse strategies: multiple search, multiple genetic operations and multiple solution pools; and (ii) `FEM` (Shen et al., 2025): an annealing-based method with additional techniques tailored for Max-Cut. For the learning-based approaches, we consider the following state-of-the-art methods: (iii) `PIGNN` (Schuetz et al., 2022): an unsupervised method for QUBO problems, which leverage physics-inspired GNNs and delivers commendable performance; (iv) `ANYCSP` (Tönshoff et al., 2023): a GNN-based reinforcement learning approach for constraint satisfaction problems, utilizing a compact graph representation and global search actions; (v) `CRA` (Ichikawa, 2024): an annealing-based method that control the convexity of objectives, with GNNs leveraged to further enhance the performance; and (vi) `ROS` (Qiu et al., 2025): a GNN-guided relax-optimize-and-sample method with fine tuning techniques, designed for Max-$k$-Cut problems. Finally, we consider the advanced commercial solver (vii) `Gurobi` 10.0.2 (Gurobi Optimization, LLC, 2024).

**Model Settings.** To enforce binarity in Problem (3), we employ the function $g(x) := x^2 - x$ throughout our `PDBO` experiments. To fully utilize GPU parallelization, we sample $B$ independent initial solutions and execute `PDBO` in parallel from each starting point, retaining the best solution across all runs. Detailed parameter configurations, including the values of $B$, $\boldsymbol{y}^0$, $\alpha$, and $\beta$, are provided in Appendix C.

**Evaluation Configuration.** All experiments are conducted with a 180-second time limit using identical hardware configuration: a 12th Gen Intel Core i9-12900K CPU and an NVIDIA GeForce RTX 3090 GPU. `PDBO` is implemented in JAX 0.6.1 (Bradbury et al., 2025).

## 4.3 COMPUTATIONAL RESULTS

### 4.3.1 RESULTS FOR MAX-CUT

The objective of the Max-Cut problem is to maximize the number of edges between disjoint vertex subsets. We evaluate `PDBO` by measuring both solution quality (number of cut edges) and computational efficiency (runtime in seconds). Our main results focus on the five largest Gset graphs (G67, G70, G72, G77, and G81), each containing over 10,000 nodes, which present significant computational challenges for state-of-the-art methods. Results for smaller graphs are provided in Appendix D.1. We observe that `PDBO` exhibits rapid convergence, and its solutions can be effectively refined by using them as initial points for `ABS2`—a method that combines multiple local search strategies and demonstrates strong performance given sufficient computation time. This hybrid approach, denoted as `PDBO+ABS2`, leverages the fast convergence of `PDBO` with the refinement capability of `ABS2`.

Table 1: Results on Gset Instances for Max-Cut

| Method | G67 (n=10k) | | G70 (n=10k) | | G72 (n=10k) | | G77 (n=14k) | | G81 (n=20k) | |
|---|---|---|---|---|---|---|---|---|---|---|
| | Obj ↑ | Time ↓ | Obj ↑ | Time ↓ | Obj ↑ | Time ↓ | Obj ↑ | Time ↓ | Obj ↑ | Time ↓ |
| PIGNN | 5538 | 23.7 | 8534 | 25.2 | 5588 | 44.5 | 7896 | 42.1 | 11078 | 157.6 |
| CRA | 5948 | 53.7 | 9240 | 51.7 | 6058 | 53.9 | 8720 | 75.9 | 12450 | 120.4 |
| ROS | 6144 | 1.5 | 8872 | 1.8 | 6148 | 1.2 | 8746 | 2.2 | 12320 | 5.2 |
| ANYCSP | 6772 | 39.9 | 9379 | 35.7 | 6816 | 36.1 | 9686 | 53.5 | 13670 | 73.5 |
| FEM | 6782 | 2.4 | 5120 | 0.2 | 6824 | 2.6 | 9688 | 4.0 | 13684 | 7.5 |
| ABS2 | 6880 | 156.3 | 9510 | 175.5 | 6932 | 172.2 | 9824 | 171.1 | 13850 | 177.1 |
| Gurobi | **6944** | 51.9 | 9514 | 133.3 | **6990** | 171.6 | **9882** | 175.1 | 13848 | 179.8 |
| PDBO | 6872 | 2.5 | **9537** | 1.9 | 6906 | 2.0 | 9812 | 2.1 | 13852 | 3.3 |
| PDBO+ABS2 | 6894 | 175.8 | **9537** | 1.9 | 6950 | 170.9 | 9840 | 5.1 | **13860** | 4.6 |

Table 1 summarized the results for all baselines. The column labeled "Obj." reports the best objective value achieved by each method within a time limit of 180 seconds. The columns labeled "Time" indicates the earliest time at which the method attains the solution corresponding to the reported objective value. Note that ↑ denotes that larger values are better, and ↓ indicates that smaller values are preferred. In summary, `PDBO` exhibits substantially greater efficiency than other baselines, and the hybrid strategy `PDBO+ABS2` achieves the best performance on G70 and G81. We also note that `Gurobi` finds the best solutions to the other three instances. It is worth noting, as highlighted in Rehfeldt et al. (2023), that recent versions of Gurobi have seen major improvements on QUBO and Max-Cut problems, likely due to the integration of specialized heuristics.

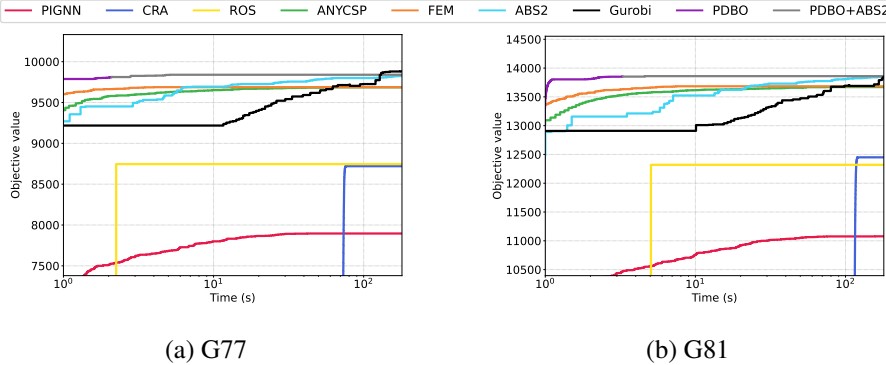

(a) G77               (b) G81

Figure 2: The objective value of Max-Cut, as a function of runtime.

Figure 2 compares the solution quality of each method throughout the optimization process, plotting the objective value against runtime under a 180-second time limit for two representative instances. Clearly, the objectives of `PDBO` (purple), continued with that of `PDBO+ABS2` (gray), remain the highest after just one second, demonstrating both high efficiency and superior solution quality. Note that while `Gurobi` (black) eventually achieves a marginally superior solution given the full time

budget, `PDBO` demonstrates significantly faster convergence, maintaining a substantial advantage for most of the optimization timeline.

### 4.3.2 RESULTS FOR MIS

The objective of the MIS problem is to find the largest set of non-adjacent nodes. Although we formulate MIS as a QUBO via constraint penalization, we only report the objective value (independent set size) for solutions that satisfy the independence constraints. We evaluate `PDBO` against baseline algorithms `PIGNN`, `CRA`, `FEM`, `ABS2` and `Gurobi` on $d$-RRG instances, with performance averaged over 20 instances per configuration, as shown in Table 2. The results show that `PIGNN`, `CRA`, `FEM` and even `Gurobi` struggle on dense graphs with degree 100 (entries marked with "–" indicate that `FEM` failed to return feasible solutions). This aligns with the known computational hardness of dense graphs (Barbier et al., 2013). In contrast, both `PDBO` and `ABS2` achieve strong performance even on dense instances. Notably, `PDBO` outperforms all baselines, obtaining better solutions in significantly less time on both sparse and dense graphs.

Table 2: Results on $d$-RRGs for MIS

| Method | $(10^4, 3)$ | | $(10^4, 100)$ | | $(5 \times 10^4, 3)$ | | $(5 \times 10^4, 100)$ | |
|---|---|---|---|---|---|---|---|---|
| | Obj $\uparrow$ | Time $\downarrow$ | Obj $\uparrow$ | Time $\downarrow$ | Obj $\uparrow$ | Time $\downarrow$ | Obj $\uparrow$ | Time $\downarrow$ |
| PIGNN | $4176.1_{\pm 19.3}$ | $66.7_{\pm 14.2}$ | $7.5_{\pm 6.7}$ | $0.2_{\pm 0.1}$ | $20705.7_{\pm 47.1}$ | $178.9_{\pm 1.0}$ | $15.8_{\pm 13.7}$ | $0.6_{\pm 0.5}$ |
| CRA | $4361.4_{\pm 8.1}$ | $52.2_{\pm 0.1}$ | $361.9_{\pm 187.6}$ | $46.6_{\pm 23.2}$ | $166.0_{\pm 75.7}$ | $20.4_{\pm 4.0}$ | $11.1_{\pm 9.8}$ | $0.5_{\pm 0.5}$ |
| FEM | $4417.2_{\pm 4.4}$ | $0.7_{\pm 0.0}$ | – | $180.0_{\pm 0.0}$ | $22053.8_{\pm 11.1}$ | $1.3_{\pm 0.1}$ | – | $180.0_{\pm 0.0}$ |
| ABS2 | $4394.8_{\pm 6.5}$ | $154.2_{\pm 27.6}$ | $587.4_{\pm 3.5}$ | $140.0_{\pm 36.2}$ | $21307.7_{\pm 43.6}$ | $173.1_{\pm 5.7}$ | $2971.5_{\pm 12.8}$ | $175.0_{\pm 3.9}$ |
| Gurobi | $4141.8_{\pm 19.4}$ | $147.2_{\pm 24.8}$ | $448.0_{\pm 5.1}$ | $0.1_{\pm 0.0}$ | $20144.3_{\pm 181.7}$ | $96.6_{\pm 49.9}$ | $2235.6_{\pm 12.7}$ | $0.1_{\pm 0.0}$ |
| PDBO | $\mathbf{4431.9}_{\pm 3.1}$ | $0.6_{\pm 0.0}$ | $\mathbf{603.0}_{\pm 3.3}$ | $3.6_{\pm 0.3}$ | $\mathbf{22127.3}_{\pm 10.3}$ | $1.1_{\pm 0.1}$ | $\mathbf{3006.2}_{\pm 6.2}$ | $14.6_{\pm 0.6}$ |

### 4.3.3 RESULTS FOR MAX-$k$-SAT.

We evaluate `PDBO` against baseline algorithms `FEM`, `ANYCSP` and `Gurobi` on Max-$k$-SAT using the 3CNF, 4CNF, and 5CNF datasets, each containing 50 problem instances. Performance is measured by the average number of unsatisfied

Table 3: Results on CNF Instances for Max-$k$-SAT

| Method | 3CNF | | 4CNF | | 5CNF | |
|---|---|---|---|---|---|---|
| | Obj $\downarrow$ | Time $\downarrow$ | Obj $\downarrow$ | Time $\downarrow$ | Obj $\downarrow$ | Time $\downarrow$ |
| FEM | $1885.2_{\pm 23.9}$ | $0.8_{\pm 0.1}$ | – | – | – | – |
| ANYCSP | $1583.3_{\pm 17.5}$ | $123.8_{\pm 40.7}$ | $\mathbf{1210.9}_{\pm 12.6}$ | $141.6_{\pm 25.9}$ | $1213.7_{\pm 11.4}$ | $141.0_{\pm 31.9}$ |
| Gurobi | $9322.6_{\pm 64.2}$ | $0.0_{\pm 0.1}$ | $9329.2_{\pm 69.7}$ | $0.0_{\pm 0.0}$ | $9310.7_{\pm 75.4}$ | $0.1_{\pm 0.0}$ |
| PDBO | $\mathbf{1583.0}_{\pm 15.7}$ | $3.5_{\pm 1.6}$ | $1230.2_{\pm 14.4}$ | $4.3_{\pm 1.8}$ | $\mathbf{1148.1}_{\pm 11.8}$ | $9.3_{\pm 3.1}$ |

clauses across all instances in each dataset, as summarized in Table 3. Entries marked with "–" indicate that `FEM` fails due to memory issues. We find that both `FEM` and `Gurobi` can hardly tackle these instances. Moreover, the results show that `PDBO` achieves competitive or superior performance compared to `ANYCSP`, while avoiding the substantial training overhead and significantly longer inference time required by the latter. We remark that future work could explore problem-specific constraint functions $g(x)$ that better align with particular Max-$k$-SAT objective landscapes, potentially improving upon the standard quadratic penalty $g(x) = x^2 - x$.

### 4.3.4 RESULTS FOR MAX-$k$-CUT

We evaluate the performance of `PDBO` against baseline algorithms `ROS`, `ANYCSP`, `FEM` and `Gurobi` on Gset instances for the Max-3-Cut problem. Following our evaluation protocol, we present results for instances with more than 10,000 nodes in Table 4, with remaining results provided in Appendix D.1. Table 4 demonstrates that `PDBO` consistently outperforms all baseline methods across all evaluated Max-3-Cut instances, achieving superior solution quality with competitive runtime efficiency.

## 5 CONCLUSIONS

In this work, we introduce PDBO, a primal-dual framework for unconstrained binary optimization that reformulates the problem as a minimax optimization and solves it with a tailored GDA algorithm. We establish convergence guarantees for PDBO and demonstrate its effectiveness through

Table 4: Results on Gset Instances for Max-3-Cut

| Method | G67 (n=10k) | | G70 (n=10k) | | G72 (n=10k) | | G77 (n=14k) | | G81 (n=20k) | |
|--------|-------------|-------------|-------------|-------------|-------------|-------------|-------------|-------------|-------------|-------------|
|        | Obj ↑ | Time ↓ | Obj ↑ | Time ↓ | Obj ↑ | Time ↓ | Obj ↑ | Time ↓ | Obj ↑ | Time ↓ |
| ROS | 7364 | 3.0 | 9983 | 1.9 | 7435 | 2.9 | 10559 | 5.3 | 14907 | 9.7 |
| ANYCSP | 7797 | 55.9 | 9909 | 16.2 | 7906 | 58.9 | 11158 | 84.0 | 15727 | 115.0 |
| FEM | 7748 | 3.3 | **9999** | 1.4 | 7835 | 0.7 | 11102 | 4.4 | 15683 | 6.1 |
| Gurobi | 7855 | 178.1 | **9999** | 0.7 | 8045 | 174.6 | 11101 | 179.9 | 15146 | 179.9 |
| PDBO | **8015** | 6.0 | **9999** | 3.3 | **8111** | 4.7 | **11467** | 5.0 | **16191** | 7.9 |

extensive experiments on public benchmarks. Our framework opens several promising directions for future research. First, while GDA provides strong empirical performance, exploring more sophisticated minimax solvers, e.g., extragradient methods, could yield better solution quality. Second, the primal-dual perspective could be extended to more complex discrete problems with additional constraints, potentially expanding its applicability to broader classes of combinatorial optimization. The demonstrated effectiveness of PDBO suggests that the reformulation of binary optimization as a minimax problem is a fruitful approach, warranting further investigation into both algorithmic improvements and theoretical understanding.

## REPRODUCIBILITY STATEMENT

To support the reproducibility of our work, we provide the following information: The source code for implementing the PDBO algorithm and reproducing the experimental results is available in an anonymous repository. The anonymous URL is provided at the beginning of Section 4. All datasets used in this study are publicly available. Detailed descriptions of the datasets are provided in Section 4.2. The hyper-parameters settings are provided in Appendix C. The complete theoretical proofs supporting our methodological claims are presented in Appendix A.

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

## THE USE OF LARGE LANGUAGE MODELS (LLMs)

In the preparation of this manuscript, we employed large language models (LLMs) solely for the purpose of language polishing and refinement. Specifically, LLM-assisted editing was used to improve grammatical accuracy and sentence fluency across the paper, particularly in sections where non-native expressions might affect readability. All essential contributions, including the basic ideas, theoretical analysis, algorithm design, experimental setup, result interpretation and conclusions, remain entirely our own.

## A  PROOFS

### A.1  PROOF OF LEMMA 1

*Proof.* By definition, for any $\boldsymbol{x} \in [0,1]^n$, $f(\boldsymbol{x})$ is the convex combination of values in $\{F(\boldsymbol{\xi}) : \boldsymbol{\xi} \in \{0,1\}^n\}$, with the weights given by $P_{\boldsymbol{x}}$. Therefore, $f(\boldsymbol{x}) \geq \min_{\boldsymbol{\xi} \in \{0,1\}^n} F(\boldsymbol{\xi})$ for any $\boldsymbol{x} \in [0,1]^n$. This means $f$ does not introduce extra minimum on $[0,1]^n \backslash \{0,1\}^n$, and hence the conclusion follows evidently.

$\square$

### A.2  PROOF OF PROPOSITION 1

*Proof.*

$$
\begin{aligned}
f(\boldsymbol{x}) &= \mathbb{E}_{\boldsymbol{\xi} \sim P_{\boldsymbol{x}}} \left( \sum_{\boldsymbol{d}} a_{\boldsymbol{d}} \xi_1^{d_1} ... \xi_n^{d_n} \right) && \text{\#by definition} \\
&= \sum_{\boldsymbol{d}} a_{\boldsymbol{d}} \left( \mathbb{E}_{\boldsymbol{\xi} \sim P_{\boldsymbol{x}}} \xi_1^{d_1} ... \xi_n^{d_n} \right) && \text{\# interchange } \mathbb{E} \text{ and } \sum \\
&= \sum_{\boldsymbol{d}} a_{\boldsymbol{d}} \left( \mathbb{E}_{\boldsymbol{\xi} \sim P_{\boldsymbol{x}}} \xi_1^{d_1} \right) \cdots \left( \mathbb{E}_{\boldsymbol{\xi} \sim P_{\boldsymbol{x}}} \xi_n^{d_n} \right) && \text{\# } \xi_i \text{ are independent} \\
&= \sum_{\boldsymbol{d}} a_{\boldsymbol{d}} \left( \mathbb{E}_{\boldsymbol{\xi} \sim P_{\boldsymbol{x}}} \xi_1 \right) \cdots \left( \mathbb{E}_{\boldsymbol{\xi} \sim P_{\boldsymbol{x}}} \xi_n \right) && \text{\# } \xi_i^{d_i} = \xi_i \text{ (for } d_i \neq 0\text{), since } \xi_i \in \{0,1\} \\
&= \sum_{\boldsymbol{d}} a_{\boldsymbol{d}} \prod_{i:d_i \neq 0} x_i && \text{\# } \mathbb{E}_{\boldsymbol{\xi} \sim P_{\boldsymbol{x}}} \xi_i = x_i
\end{aligned}
$$

$\square$

### A.3  PROOF OF PROPOSITION 2

*Proof.* Observe that

$$
\begin{aligned}
\sup_{\boldsymbol{y} \in \mathbb{R}^n} L(\boldsymbol{x}, \boldsymbol{y}) &= f(\boldsymbol{x}) + \sup_{\boldsymbol{y} \in \mathbb{R}^n} \sum_{i=1}^n y_i g(x_i) \\
&= \begin{cases} f(\boldsymbol{x}), & \text{if } \boldsymbol{x} \in \{0,1\}^n \\ +\infty, & \text{otherwise} \end{cases}
\end{aligned}
$$

Therefore, $\inf_{\boldsymbol{x} \in [0,1]^n} \sup_{\boldsymbol{y} \in \mathbb{R}^n} L(\boldsymbol{x}, \boldsymbol{y}) = \inf_{\boldsymbol{x} \in \{0,1\}^n} f(\boldsymbol{x}) = v^*$

$\square$

### A.4  PROOF OF PROPOSITION 3

*Proof.* The three conditions for $g$ derive the following properties:

- $g(\frac{1}{2}) \leq g(x) < 0$ for any $x \in (0,1)$

- $g'(x)$ is monotone non-decreasing

- $g'(\frac{1}{2} - \delta) < 0 < g'(\frac{1}{2} + \delta)$ and $g'(\frac{1}{2} - \delta) + g'(\frac{1}{2} + \delta) = 0$

For any $t \geq 0$, the update $y_i^{t+1} - y_i^t = \beta \cdot g(x_i^t)$ lies within $[\beta \cdot g(\frac{1}{2}), 0]$. Consequently, the sequence $\{y_i^t\}_{t \geq 0}$ is monotone non-increasing. If $y_i^t > \frac{1+\Theta}{g'(\frac{1}{2}-\delta)}$ for any $t \geq 0$, there is nothing left to prove, otherwise we denote by $T$ the smallest time index such that $y_i^T \leq \frac{1+\Theta}{g'(\frac{1}{2}-\delta)}$ and $x_i^T \notin [\frac{1}{2} - \delta, \frac{1}{2} + \delta]$. We have $y_i^T \in [2\beta \cdot g(x_i^t) + \frac{1+\Theta}{g'(\frac{1}{2}-\delta)}, \frac{1+\Theta}{g'(\frac{1}{2}-\delta)}]$, since Algorithm 1 prevents the stagnation of $x_i^t$ within $[0, 1]$ for any two successive steps. Now we discuss if $x_i^T$ lies in $[0, \frac{1}{2} - \delta]$ or $[\frac{1}{2} + \delta, 1]$.

**Case I.** If $x_i^T \in [0, \frac{1}{2} - \delta]$, then $g'(x_i^T) \leq g'(\frac{1}{2} - \delta) < 0$, and hence

$$\frac{\partial}{\partial x_i} L(\boldsymbol{x}^T, \boldsymbol{y}^T) = \frac{\partial}{\partial x_i} f(\boldsymbol{x}^T) + y_i^T \cdot g'(x_i^T)$$

$$\geq -\Theta + \frac{1+\Theta}{g'(\frac{1}{2}-\delta)} \cdot g'(\frac{1}{2} - \delta)$$

$$= 1$$

For the primal update

$$x_i^{T+1} = \Pi_{[0,1]} \left[ x_i^T - \alpha \cdot \frac{\partial}{\partial x_i} L(\boldsymbol{x}^T, \boldsymbol{y}^T) \right]$$

$$\leq \Pi_{[0,1]} \left[ x_i^T - \alpha \right]$$

$$\leq x_i^T$$

Using an analogous argument, we induce that the sequence $\{x_i^{T+t}\}_{t \geq 0}$ is non-increasing and decreases at least $\alpha$ in each step until it reaches 0 after at most $\frac{\frac{1}{2}-\delta}{\alpha} \leq \lceil \frac{1}{2\alpha} \rceil$ steps. Therefore, we conclude that both $x_i^t$ and $y_i^t$ remain unchanged for $t \geq T + \lceil \frac{1}{2\alpha} \rceil$, and

$$y_i^{T+\lceil \frac{1}{2\alpha} \rceil} = y_i^T + \sum_{t=1}^{\lceil \frac{1}{2\alpha} \rceil} \left( y_i^{T+t} - y_i^{T+t-1} \right)$$

$$\geq \left( 2\beta \cdot g(x_i^T) + \frac{1+\Theta}{g'(\frac{1}{2}-\delta)} \right) + \left( \lceil \frac{1}{2\alpha} \rceil \cdot \beta \cdot g(\frac{1}{2}) \right)$$

$$\geq \frac{1+\Theta}{g'(\frac{1}{2}-\delta)} + \left( 2 + \lceil \frac{1}{2\alpha} \rceil \right) \cdot \beta \cdot g(\frac{1}{2}) \qquad \# g(x_i^T) \geq g(\frac{1}{2})$$

**Case II.** If $x_i^T \in [\frac{1}{2} + \delta, 1]$, then $g'(x_i^T) \geq g'(\frac{1}{2} + \delta) > 0$, and hence

$$\frac{\partial}{\partial x_i} L(\boldsymbol{x}^t, \boldsymbol{y}^t) = \frac{\partial}{\partial x_i} f(\boldsymbol{x}^t) + y_i^T \cdot g'(x_i^T)$$

$$\leq \Theta + \frac{1+\Theta}{g'(\frac{1}{2}-\delta)} \cdot g'(\frac{1}{2} + \delta)$$

$$= -1$$

For the primal update

$$x_i^{T+1} = \Pi_{[0,1]} \left[ x_i^T - \alpha \cdot \frac{\partial}{\partial x_i} L(\boldsymbol{x}^t, \boldsymbol{y}^t) \right]$$

$$\geq \Pi_{[0,1]} \left[ x_i^T + \alpha \right]$$

$$\geq x_i^T$$

Using an analogous argument, we induce that the sequence $\{x_i^{T+t}\}_{t \geq 0}$ is non-decreasing and increases at least $\alpha$ in each step until it reaches 1 after at most $\frac{\frac{1}{2}-\delta}{\alpha} \leq \lceil \frac{1}{2\alpha} \rceil$ steps.

Therefore, we conclude that both $x_i^t$ and $y_i^t$ remain unchanged for $t \geq T + \lceil \frac{1}{2\alpha} \rceil$, and

$$
\begin{aligned}
y_i^{T+\lceil \frac{1}{2\alpha} \rceil} &= y_i^T + \sum_{t=1}^{\lceil \frac{1}{2\alpha} \rceil} \left( y_i^{T+t} - y_i^{T+t-1} \right) \\
&\geq \left( 2\beta \cdot g(x_i^T) + \frac{1+\Theta}{g'(\frac{1}{2} - \delta)} \right) + \left( \lceil \frac{1}{2\alpha} \rceil \cdot \beta \cdot g(\frac{1}{2}) \right) \\
&\geq \frac{1+\Theta}{g'(\frac{1}{2} - \delta)} + \left( 2 + \lceil \frac{1}{2\alpha} \rceil \right) \cdot \beta \cdot g(\frac{1}{2})
\end{aligned}
$$

$\square$

## A.5 PROOF OF THEOREM 1

*Proof.* Clearly, we have the following weak duality:

$$
\inf_{\boldsymbol{x} \in [0,1]^n} \sup_{\boldsymbol{y} \in \mathbb{R}^n} L(\boldsymbol{x}, \boldsymbol{y}) \geq \sup_{\boldsymbol{y} \in \mathbb{R}^n} \inf_{\boldsymbol{x} \in [0,1]^n} L(\boldsymbol{x}, \boldsymbol{y}).
$$

Taking $\bar{\boldsymbol{y}} := \boldsymbol{0}$ yields $\inf_{\boldsymbol{x} \in [0,1]^n} L(\boldsymbol{x}, \bar{\boldsymbol{y}}) = \inf_{\boldsymbol{x} \in [0,1]^n} f(\boldsymbol{x})$. Therefore, we have

$$
\begin{aligned}
\sup_{\boldsymbol{y} \in \mathbb{R}^n} \inf_{\boldsymbol{x} \in [0,1]^n} L(\boldsymbol{x}, \boldsymbol{y}) &\geq \inf_{\boldsymbol{x} \in [0,1]^n} L(\boldsymbol{x}, \bar{\boldsymbol{y}}) \\
&= \inf_{\boldsymbol{x} \in [0,1]^n} f(\boldsymbol{x}) \\
&= \min_{\boldsymbol{x} \in \{0,1\}^n} f(\boldsymbol{x}) \qquad \text{(Lemma 1)} \\
&= \inf_{\boldsymbol{x} \in [0,1]^n} \sup_{\boldsymbol{y} \in \mathbb{R}^n} L(\boldsymbol{x}, \boldsymbol{y}) \quad \text{(Proposition 2)}
\end{aligned}
$$

$\square$

## A.6 PROOF OF THEOREM 2

*Proof.* Given the update $y_i^{t+1} = y_i^t + \beta \cdot g(x_i^t)$, we have $\sum_{t=0}^{T-1} -\beta \cdot g(x_i^t) = y_i^0 - y_i^T \leq y_i^0 - y_i^*$ for any $i \in [n]$ and any $T \geq 1$. Consequently,

$$
\frac{\sum_{t=0}^{T-1} \sum_{i=1}^{n} -g(x_i^t)}{T} \leq \frac{||\boldsymbol{y}^0 - \boldsymbol{y}^*||_1}{\beta \cdot T}
$$

Therefore, the number of Algorithm 1 to return an $\epsilon$-binary point is bounded by $\mathcal{O}\left( \frac{||\boldsymbol{y}^0 - \boldsymbol{y}^*||_1}{\beta \epsilon} \right)$. $\square$

# B GENERALIZATION TO THE MAX-$k$-CUT PROBLEM

Given a graph with $n$ nodes, the Max-$k$-Cut problem aims to partition the nodes into $k$ categories that maximize the number of edges connecting the nodes from distinct categories:

$$
\begin{aligned}
\max_{\boldsymbol{X} \in \mathbb{R}^{k \times n}} \quad & \frac{1}{2} \sum_{i=1}^{n} \sum_{j=1}^{n} W_{i,j} \left( 1 - \boldsymbol{X}_{:,i}^\top \boldsymbol{X}_{:,j} \right) \\
\text{s.t.} \quad & \boldsymbol{X}_{:,i} \in \left\{ \boldsymbol{e}^{(1)}, \boldsymbol{e}^{(2)}, ..., \boldsymbol{e}^{(k)} \right\}, \quad \forall i \in [n]
\end{aligned}
$$

where $\boldsymbol{W} \in \mathbb{R}^{n \times n}$ is a symmetric matrix representing the weights of edges and each node $i$ is assigned with a one-hot vector $\boldsymbol{X}_{:,i}$ representing its category. The formulation can be equivalently expressed as:

$$
\begin{aligned}
\min_{\boldsymbol{X} \in \mathbb{R}^{k \times n}} \quad & \text{Tr}\left( \boldsymbol{X} \boldsymbol{W} \boldsymbol{X}^\top \right) \\
\text{s.t.} \quad & \boldsymbol{X}_{:,i} \in \left\{ \boldsymbol{e}^{(1)}, \boldsymbol{e}^{(2)}, ..., \boldsymbol{e}^{(k)} \right\}, \quad \forall i \in [n]
\end{aligned}
$$

Note that the domain of each $\boldsymbol{X}_{:,i}$ can be equivalently expressed as:

$$\left\{ \boldsymbol{e}^{(1)}, \boldsymbol{e}^{(2)}, ..., \boldsymbol{e}^{(k)} \right\} = \Delta_k \cap \left\{ \boldsymbol{X}_{:,i} \in \mathbb{R}^k : g(\boldsymbol{X}_{:,i}) = 0 \right\}$$

where $\Delta_k := \{\boldsymbol{x} \in \mathbb{R}_+^k : \sum_{i=1}^{k} x_i = 1\}$ denotes the $k$-dimensional probability simplex, and $g$ is the convex constraint function as described in Section 3.1. We summarize two possible construction of $g$:

- $g(\boldsymbol{X}_{:,i}) := \sum_{j=1}^{k} \boldsymbol{X}_{j,i}^2 - 1$            (quadratic)

- $g(\boldsymbol{X}_{:,i}) := \sum_{j=1}^{k} \boldsymbol{X}_{j,i} \log(\boldsymbol{X}_{j,i})$        (entropy)

We can now reformulate the Max-$k$-Cut problem as:

$$\min_{\boldsymbol{X} \in \Delta_k^n} \quad \mathrm{Tr}\left(\boldsymbol{X}\boldsymbol{W}\boldsymbol{X}^\top\right)$$
$$\text{s.t.} \quad g(\boldsymbol{X}_{:,i}) = 0 \quad \forall i \in [n]$$

The Lagrangian is defined as:

$$L(\boldsymbol{X}, \boldsymbol{y}) := \mathrm{Tr}\left(\boldsymbol{X}\boldsymbol{W}\boldsymbol{X}^\top\right) + \sum_{i=1}^{n} y_i g(\boldsymbol{X}_{:,i})$$

To optimize the corresponding minimax problem $\min_{\boldsymbol{X} \in \Delta_k^n} \max_{\boldsymbol{y} \in \mathbb{R}^n} L(\boldsymbol{X}, \boldsymbol{y})$, a straightforward approach is to align with Algorithm 1 by projecting $\boldsymbol{X}^t$ onto $\Delta_k^n$. While there are a variety of methods for normalization, we adopt an alternative by re-parameterizing $\boldsymbol{X}_{:,i} \in \Delta_k$ via the softmax function:

$$\boldsymbol{X}_{:,i} = \mathrm{SoftMax}(\boldsymbol{Z}_{:,i}) = \left( \frac{\exp(\boldsymbol{Z}_{j,i})}{\sum_{j=1}^{k} \exp(\boldsymbol{Z}_{j,i})} \right)_{j=1,...,k}$$

and the minimax problem is converted to

$$\min_{\boldsymbol{Z} \in \mathbb{R}^{k \times n}} \max_{\boldsymbol{y} \in \mathbb{R}^n} \quad L(\mathrm{SoftMax}(\boldsymbol{Z}), \boldsymbol{y})$$

Consequently, we can apply the approach described in Section 3.2 to the Max-$k$-Cut problem.

## C    HYPER-PARAMETERS

The multiple initial solutions are selected as follows: The initial dual variables is set to $\boldsymbol{y}^0 := \bar{y} \cdot \mathbf{1}$ and the initial primal variable is sampled from the uniform distribution $\boldsymbol{x}^0 \sim \mathrm{Uniform}([0,1]^n)$. The hyper-parameters can be efficiently found via a grid search on small-scale problems. Table 5 summarizes the hyper-parameters employed in our experiments.

## D    ADDITIONAL EXPERIMENTAL RESULTS

### D.1    COMPLETE RESULTS ON GSET INSTANCES FOR MAX-CUT AND MAX-k-CUT

We provide complete results on the Gset instances. Table 6 exhibits the complete results on Gset instances for the Max-Cut problem, while Table 7 exhibits the complete results on Gset instances for the Max-3-Cut problem.

Table 5: Hyper-parameters

| Hyper-parameter | Notation | Max-Cut | MIS | Max-$k$-SAT | Max-$k$-Cut |
|---|---|---|---|---|---|
| Number of initial solutions | $B$ | 100 | 10 | 10 | 100 |
| Initial dual variable | $\bar{y}$ | 6 | 5 | 2 | 6 |
| Primal step size | $\alpha$ | 0.025 | 0.02 | 0.01 | 0.01 |
| Dual step size | $\beta$ | 0.025 | 0.02 | 0.005 | 0.01 |

While PDBO consistently delivers superior performance on Max-3-Cut, the highly specialized ABS2 solver is a strong performer on small-scale Max-Cut instances. On the 66 benchmarks with up to 10,000 nodes, PDBO alone is competitive, matching or beating ABS2 in 16 cases. The most effective strategy, however, is a hybrid approach: using PDBO for rapid initialization followed by ABS2's intensive local search. This hybrid method matches or surpasses standalone ABS2 on 44 of the 66 instances, demonstrating a valuable synergy.

Moreover, note that ABS2 is not applicable to Max-k-Cut and is confined to QUBO/Max-Cut. In contrast, PDBO's first-order optimization strategy allows it to address a broader class of combinatorial problems with significant efficiency.

Table 6: Computational results on Gset instances for Max-Cut.

| Instance | $|\mathcal{V}|$ | $|\mathcal{E}|$ | PIGNN Obj. ↑ | PIGNN Time (s) ↓ | CRA Obj. ↑ | CRA Time (s) ↓ | ROS Obj. ↑ | ROS Time (s) ↓ | ANYCSP Obj. ↑ | ANYCSP Time (s) ↓ | FEM Obj. ↑ | FEM Time (s) ↓ | ABS2 Obj. ↑ | ABS2 Time (s) ↓ | PDBO Obj. ↑ | PDBO Time (s) ↓ | PDBO+ABS2 Obj. ↑ | PDBO+ABS2 Time (s) ↓ |
|---|---|---|---|---|---|---|---|---|---|---|---|---|---|---|---|---|---|---|
| G1 | 800 | 19176 | 10891 | 14.3 | 11398 | 25.0 | 11313 | 0.8 | 11623 | 6.5 | 11624 | 0.8 | 11624 | 0.1 | 11624 | 1.6 | 11624 | 1.6 |
| G2 | 800 | 19176 | 10930 | 23.3 | 11498 | 23.7 | 11417 | 1.1 | 11612 | 10.5 | 11617 | 0.8 | 11620 | 0.2 | 11617 | 1.4 | 11620 | 3.6 |
| G3 | 800 | 19176 | 10958 | 0.2 | 11464 | 21.5 | 11383 | 0.7 | 11620 | 3.8 | 11622 | 0.8 | 11622 | 0.2 | 11621 | 1.0 | 11622 | 1.2 |
| G4 | 800 | 19176 | 10973 | 0.7 | 11486 | 24.0 | 11357 | 0.6 | 11644 | 9.8 | 11646 | 0.7 | 11646 | 1.1 | 11646 | 0.9 | 11646 | 0.9 |
| G5 | 800 | 19176 | 10901 | 0.2 | 11463 | 24.8 | 11410 | 0.8 | 11630 | 7.3 | 11627 | 0.7 | 11631 | 0.1 | 11631 | 1.7 | 11631 | 1.7 |
| G6 | 800 | 19176 | 1751 | 33.3 | 1747 | 46.7 | 1949 | 0.9 | 2173 | 13.5 | 2178 | 0.8 | 2178 | 0.1 | 2178 | 1.4 | 2178 | 1.4 |
| G7 | 800 | 19176 | 1548 | 19.0 | 1512 | 48.1 | 1772 | 0.9 | 1997 | 3.7 | 2001 | 0.8 | 2006 | 7.1 | 2001 | 2.2 | 2001 | 2.2 |
| G8 | 800 | 19176 | 1588 | 58.5 | 1490 | 48.1 | 1713 | 1.1 | 2004 | 10.2 | 2004 | 0.8 | 2005 | 0.2 | 2003 | 1.3 | 2005 | 5.6 |
| G9 | 800 | 19176 | 1621 | 26.8 | 1544 | 46.8 | 1784 | 0.8 | 2044 | 13.5 | 2051 | 0.8 | 2054 | 0.1 | 2054 | 1.4 | 2054 | 1.4 |
| G10 | 800 | 19176 | 1635 | 21.0 | 1481 | 48.1 | 1755 | 1.3 | 1991 | 4.8 | 1999 | 0.8 | 2000 | 3.0 | 1999 | 0.9 | 1999 | 0.9 |
| G11 | 800 | 1600 | 478 | 3.1 | 440 | 45.2 | 492 | 0.7 | 558 | 1.4 | 558 | 0.3 | 564 | 0.1 | 562 | 0.6 | 564 | 1.7 |
| G12 | 800 | 1600 | 472 | 3.6 | 428 | 44.7 | 476 | 0.4 | 548 | 0.3 | 552 | 0.4 | 556 | 0.5 | 554 | 0.5 | 554 | 0.5 |
| G13 | 800 | 1600 | 490 | 4.6 | 430 | 44.2 | 514 | 0.6 | 572 | 0.4 | 576 | 0.3 | 582 | 0.5 | 582 | 0.5 | 582 | 0.5 |
| G14 | 800 | 4694 | 2815 | 12.3 | 2987 | 39.2 | 2941 | 0.7 | 3058 | 6.2 | 3058 | 0.5 | 3064 | 2.1 | 3054 | 0.8 | 3064 | 22.9 |
| G15 | 800 | 4661 | 2746 | 3.8 | 2951 | 38.4 | 2944 | 0.7 | 3040 | 2.8 | 3047 | 0.5 | 3045 | 1.2 | 3041 | 0.9 | 3043 | 2.6 |
| G16 | 800 | 4672 | 2735 | 1.8 | 2966 | 39.5 | 2955 | 0.8 | 3040 | 3.4 | 3049 | 0.5 | 3051 | 1.2 | 3044 | 0.9 | 3046 | 79.9 |
| G17 | 800 | 4667 | 2699 | 40.8 | 2948 | 39.2 | 2952 | 0.8 | 3037 | 2.2 | 3043 | 0.5 | 3042 | 63.6 | 3037 | 0.8 | 3041 | 8.5 |
| G18 | 800 | 4694 | 833 | 14.9 | 822 | 44.9 | 859 | 1.1 | 989 | 4.2 | 991 | 0.5 | 991 | 33.0 | 987 | 0.8 | 992 | 8.1 |
| G19 | 800 | 4661 | 721 | 19.9 | 771 | 45.8 | 776 | 0.8 | 899 | 1.5 | 905 | 0.5 | 906 | 0.7 | 902 | 0.7 | 902 | 0.7 |
| G20 | 800 | 4672 | 668 | 7.7 | 771 | 44.3 | 838 | 1.2 | 936 | 1.7 | 941 | 0.5 | 941 | 0.1 | 940 | 1.0 | 941 | 1.2 |
| G21 | 800 | 4667 | 753 | 11.1 | 766 | 46.7 | 810 | 0.5 | 925 | 5.7 | 930 | 0.4 | 931 | 1.1 | 924 | 0.7 | 931 | 2.5 |
| G22 | 2000 | 19990 | 12308 | 13.1 | 13116 | 35.4 | 12920 | 1.0 | 13347 | 13.9 | 13356 | 0.8 | 13357 | 10.4 | 13347 | 2.4 | 13359 | 4.8 |
| G23 | 2000 | 19990 | 12288 | 2.1 | 13109 | 35.0 | 12996 | 1.4 | 13328 | 7.5 | 13338 | 0.8 | 13334 | 1.4 | 13327 | 2.3 | 13337 | 41.6 |
| G24 | 2000 | 19990 | 12283 | 12.1 | 13067 | 35.8 | 12935 | 1.1 | 13316 | 9.9 | 13329 | 0.8 | 13327 | 32.9 | 13317 | 2.4 | 13317 | 2.4 |
| G25 | 2000 | 19990 | 12251 | 12.6 | 13122 | 36.2 | 13008 | 0.6 | 13323 | 12.9 | 13334 | 0.8 | 13329 | 55.2 | 13321 | 2.2 | 13325 | 18.3 |
| G26 | 2000 | 19990 | 12247 | 14.4 | 13028 | 36.4 | 12952 | 0.8 | 13311 | 10.8 | 13319 | 0.8 | 13314 | 4.7 | 13309 | 2.3 | 13315 | 4.1 |
| G27 | 2000 | 19990 | 2576 | 57.2 | 2749 | 46.6 | 2895 | 1.0 | 3314 | 16.2 | 3339 | 0.8 | 3331 | 8.0 | 3326 | 1.1 | 3326 | 1.1 |
| G28 | 2000 | 19990 | 2581 | 122.1 | 2685 | 46.6 | 2972 | 0.8 | 3275 | 18.1 | 3291 | 0.8 | 3283 | 1.5 | 3296 | 2.1 | 3296 | 2.1 |
| G29 | 2000 | 19990 | 2730 | 50.6 | 2763 | 46.6 | 2991 | 1.1 | 3381 | 17.6 | 3396 | 0.8 | 3391 | 36.4 | 3383 | 2.4 | 3404 | 23.9 |
| G30 | 2000 | 19990 | 2675 | 130.2 | 2748 | 46.8 | 2951 | 1.0 | 3399 | 6.2 | 3409 | 0.8 | 3411 | 0.9 | 3402 | 1.9 | 3402 | 1.9 |
| G31 | 2000 | 19990 | 2601 | 40.4 | 2641 | 47.6 | 2890 | 0.7 | 3296 | 16.3 | 3302 | 0.8 | 3288 | 1.0 | 3302 | 2.3 | 3306 | 5.6 |
| G32 | 2000 | 4000 | 1170 | 5.9 | 1156 | 44.7 | 1234 | 1.3 | 1386 | 8.7 | 1388 | 0.5 | 1410 | 72.6 | 1398 | 0.6 | 1410 | 61.7 |
| G33 | 2000 | 4000 | 1134 | 12.0 | 1122 | 43.8 | 1220 | 0.7 | 1354 | 3.3 | 1360 | 0.4 | 1382 | 52.7 | 1370 | 0.6 | 1380 | 94.5 |
| G34 | 2000 | 4000 | 1122 | 14.4 | 1124 | 44.8 | 1204 | 0.5 | 1358 | 7.0 | 1358 | 0.4 | 1384 | 9.9 | 1376 | 0.9 | 1384 | 127.6 |
| G35 | 2000 | 11778 | 6968 | 4.5 | 7453 | 39.4 | 7460 | 0.7 | 7650 | 8.0 | 7666 | 0.7 | 7661 | 12.1 | 7643 | 1.2 | 7660 | 8.2 |

Table 6 continued.

| Instance | $|\mathcal{V}|$ | $|\mathcal{E}|$ | PIGNN Obj. ↑ | PIGNN Time (s) ↓ | CRA Obj. ↑ | CRA Time (s) ↓ | ROS Obj. ↑ | ROS Time (s) ↓ | ANYCSP Obj. ↑ | ANYCSP Time (s) ↓ | FEM Obj. ↑ | FEM Time (s) ↓ | ABS2 Obj. ↑ | ABS2 Time (s) ↓ | PDBO Obj. ↑ | PDBO Time (s) ↓ | PDBO+ABS2 Obj. ↑ | PDBO+ABS2 Time (s) ↓ |
|---|---|---|---|---|---|---|---|---|---|---|---|---|---|---|---|---|---|---|
| G36 | 2000 | 11766 | 6967 | 5.7 | 7435 | 38.9 | 7423 | 1.4 | 7644 | 11.7 | 7659 | 0.7 | 7666 | 31.4 | 7636 | 1.0 | 7646 | 5.6 |
| G37 | 2000 | 11785 | 7006 | 6.7 | 7466 | 39.9 | 7432 | 1.2 | 7654 | 8.9 | 7670 | 0.7 | 7667 | 56.8 | 7657 | 0.9 | 7676 | 54.1 |
| G38 | 2000 | 11779 | 6993 | 1.3 | 7450 | 39.1 | 7468 | 1.8 | 7652 | 7.6 | 7664 | 0.7 | 7672 | 14.2 | 7654 | 1.5 | 7665 | 13.6 |
| G39 | 2000 | 11778 | 1961 | 12.3 | 2115 | 46.8 | 2113 | 1.2 | 2384 | 6.8 | 2396 | 0.6 | 2385 | 1.4 | 2387 | 2.0 | 2387 | 2.0 |
| G40 | 2000 | 11766 | 1915 | 104.4 | 2112 | 44.5 | 2149 | 1.6 | 2376 | 13.9 | 2394 | 0.7 | 2389 | 3.7 | 2365 | 1.4 | 2396 | 83.8 |
| G41 | 2000 | 11785 | 1976 | 21.1 | 2108 | 46.0 | 2090 | 1.8 | 2371 | 5.3 | 2391 | 0.6 | 2405 | 33.4 | 2369 | 1.5 | 2376 | 3.3 |
| G42 | 2000 | 11779 | 1972 | 31.3 | 2203 | 44.4 | 2191 | 0.6 | 2462 | 3.8 | 2464 | 0.7 | 2468 | 2.4 | 2450 | 1.3 | 2476 | 87.8 |
| G43 | 1000 | 9990 | 6210 | 22.9 | 6537 | 33.8 | 6427 | 0.8 | 6660 | 7.3 | 6660 | 0.6 | 6660 | 0.1 | 6660 | 1.2 | 6660 | 1.2 |
| G44 | 1000 | 9990 | 6160 | 1.0 | 6546 | 33.9 | 6433 | 0.6 | 6646 | 3.3 | 6650 | 0.6 | 6650 | 3.3 | 6650 | 1.3 | 6650 | 1.3 |
| G45 | 1000 | 9990 | 6152 | 1.7 | 6520 | 35.4 | 6473 | 0.8 | 6646 | 2.0 | 6653 | 0.6 | 6654 | 7.3 | 6653 | 1.1 | 6653 | 1.1 |
| G46 | 1000 | 9990 | 6122 | 2.3 | 6564 | 34.1 | 6463 | 0.8 | 6639 | 7.7 | 6646 | 0.6 | 6649 | 2.0 | 6649 | 1.0 | 6649 | 1.0 |
| G47 | 1000 | 9990 | 6174 | 1.6 | 6503 | 35.0 | 6487 | 0.8 | 6655 | 8.9 | 6656 | 0.6 | 6657 | 3.2 | 6656 | 1.6 | 6657 | 11.2 |
| G48 | 3000 | 6000 | 5072 | 23.1 | 5852 | 41.9 | 5624 | 0.9 | 6000 | 1.5 | 6000 | 0.3 | 6000 | 0.2 | 6000 | 0.4 | 6000 | 0.4 |
| G49 | 3000 | 6000 | 4992 | 16.0 | 5898 | 42.1 | 5552 | 0.9 | 6000 | 0.9 | 6000 | 0.3 | 6000 | 0.3 | 6000 | 0.4 | 6000 | 0.4 |
| G50 | 3000 | 6000 | 5040 | 16.7 | 5756 | 42.7 | 5568 | 0.8 | 5878 | 2.1 | 5880 | 0.4 | 5880 | 0.4 | 5880 | 1.1 | 5880 | 1.1 |
| G51 | 1000 | 5909 | 3407 | 2.3 | 3732 | 39.3 | 3711 | 1.0 | 3832 | 5.8 | 3843 | 0.5 | 3845 | 31.3 | 3834 | 0.9 | 3841 | 5.5 |
| G52 | 1000 | 5916 | 3458 | 1.1 | 3742 | 39.2 | 3734 | 1.7 | 3837 | 7.0 | 3845 | 0.6 | 3846 | 70.9 | 3834 | 0.8 | 3841 | 4.8 |
| G53 | 1000 | 5914 | 3526 | 9.3 | 3739 | 39.3 | 3736 | 1.5 | 3836 | 2.5 | 3842 | 0.6 | 3847 | 31.5 | 3838 | 0.9 | 3840 | 2.2 |
| G54 | 1000 | 5916 | 3314 | 2.4 | 3743 | 39.4 | 3722 | 1.4 | 3838 | 7.2 | 3844 | 0.5 | 3845 | 35.2 | 3836 | 1.1 | 3851 | 174.6 |
| G55 | 5000 | 12498 | 9059 | 84.3 | 10008 | 43.1 | 9797 | 1.4 | 10248 | 22.6 | 6418 | 0.2 | 10254 | 93.5 | 10240 | 1.3 | 10258 | 120.3 |
| G56 | 5000 | 12498 | 3127 | 106.0 | 3414 | 47.1 | 3439 | 0.9 | 3978 | 13.7 | 127 | 0.2 | 3970 | 63.0 | 3956 | 1.5 | 3970 | 136.7 |
| G57 | 5000 | 10000 | 2794 | 18.5 | 2942 | 45.9 | 3036 | 0.9 | 3398 | 18.0 | 3408 | 0.8 | 3474 | 150.0 | 3460 | 1.2 | 3478 | 157.7 |
| G58 | 5000 | 29570 | 17435 | 7.8 | 18695 | 41.8 | 18692 | 1.4 | 19187 | 29.7 | 19211 | 1.3 | 19200 | 120.2 | 19163 | 3.6 | 19192 | 180.2 |
| G59 | 5000 | 29570 | 4734 | 170.8 | 5580 | 48.6 | 5298 | 0.9 | 5978 | 33.1 | 6036 | 1.4 | 6007 | 51.8 | 5967 | 2.7 | 6006 | 98.2 |
| G60 | 7000 | 17148 | 12382 | 32.4 | 13764 | 46.1 | 13413 | 1.0 | 14108 | 27.3 | 8746 | 0.2 | 14138 | 171.1 | 14111 | 2.0 | 14111 | 2.0 |
| G61 | 7000 | 17148 | 4518 | 58.0 | 4951 | 49.4 | 5061 | 1.0 | 5736 | 19.9 | 326 | 0.2 | 5707 | 30.7 | 5717 | 1.8 | 5717 | 1.8 |
| G62 | 7000 | 14000 | 3878 | 16.0 | 4108 | 48.6 | 4268 | 1.3 | 4758 | 13.6 | 4762 | 1.1 | 4848 | 176.5 | 4808 | 1.8 | 4828 | 175.2 |
| G63 | 7000 | 41459 | 24237 | 106.0 | 26183 | 43.5 | 26209 | 2.3 | 26844 | 48.0 | 26941 | 2.0 | 26893 | 179.4 | 26881 | 3.9 | 26881 | 3.9 |
| G64 | 7000 | 41459 | 6829 | 105.0 | 8114 | 50.3 | 7665 | 1.3 | 8599 | 45.9 | 8660 | 1.9 | 8622 | 14.9 | 8591 | 4.5 | 8591 | 4.5 |
| G65 | 8000 | 16000 | 4446 | 35.2 | 4814 | 50.9 | 4898 | 1.0 | 5414 | 26.9 | 5420 | 1.6 | 5520 | 148.3 | 5482 | 1.6 | 5504 | 167.9 |
| G66 | 9000 | 18000 | 5142 | 20.9 | 5488 | 51.7 | 5598 | 1.5 | 6192 | 38.2 | 6196 | 2.0 | 6302 | 165.9 | 6268 | 1.8 | 6318 | 160.2 |
| G67 | 10000 | 20000 | 5538 | 23.7 | 5948 | 53.7 | 6144 | 1.5 | 6772 | 39.9 | 6782 | 2.4 | 6880 | 156.3 | 6872 | 2.5 | 6894 | 175.8 |
| G70 | 10000 | 9999 | 8534 | 25.2 | 9240 | 51.7 | 8872 | 1.8 | 9379 | 35.7 | 5120 | 0.2 | 9510 | 175.5 | 9537 | 1.9 | 9537 | 1.9 |
| G72 | 10000 | 20000 | 5588 | 44.5 | 6058 | 53.9 | 6148 | 1.2 | 6816 | 36.1 | 6824 | 2.6 | 6932 | 172.2 | 6906 | 2.0 | 6950 | 170.9 |
| G77 | 14000 | 28000 | 7896 | 42.1 | 8720 | 75.9 | 8746 | 2.2 | 9686 | 53.5 | 9688 | 4.0 | 9824 | 171.1 | 9812 | 2.1 | 9840 | 5.1 |
| G81 | 20000 | 40000 | 11078 | 157.6 | 12450 | 120.4 | 12320 | 5.2 | 13670 | 73.5 | 13684 | 7.5 | 13850 | 177.1 | 13852 | 3.3 | 13860 | 4.6 |

Table 7: Computational results on Gset instances for Max-3-Cut.

| Instance | $|\mathcal{V}|$ | $|\mathcal{E}|$ | ROS | | ANYCSP | | FEM | | PDBO | |
|---|---|---|---|---|---|---|---|---|---|---|
| | | | Obj. ↑ | Time (s) ↓ | Obj. ↑ | Time (s) ↓ | Obj. ↑ | Time (s) ↓ | Obj. ↑ | Time (s) ↓ |
| G1 | 800 | 19176 | 14892 | 1.2 | 15121 | 18.0 | 15058 | 1.0 | 15144 | 3.1 |
| G2 | 800 | 19176 | 14892 | 1.3 | 15119 | 15.8 | 15070 | 0.8 | 15159 | 3.7 |
| G3 | 800 | 19176 | 14932 | 1.2 | 15129 | 17.0 | 15080 | 1.1 | 15161 | 4.2 |
| G4 | 800 | 19176 | 14883 | 1.3 | 15141 | 15.7 | 15080 | 0.9 | 15182 | 4.5 |
| G5 | 800 | 19176 | 14859 | 1.4 | 15144 | 16.8 | 15087 | 1.1 | 15185 | 4.6 |
| G6 | 800 | 19176 | 2341 | 1.2 | 1364 | 10.5 | 2509 | 0.8 | 2616 | 2.9 |
| G7 | 800 | 19176 | 2152 | 1.5 | 1131 | 0.2 | 2306 | 0.8 | 2393 | 3.9 |
| G8 | 800 | 19176 | 2083 | 1.6 | 1114 | 0.2 | 2321 | 1.1 | 2413 | 3.6 |
| G9 | 800 | 19176 | 2214 | 1.5 | 1264 | 0.2 | 2357 | 0.8 | 2449 | 3.2 |
| G10 | 800 | 19176 | 2104 | 1.4 | 1110 | 0.2 | 2297 | 0.9 | 2393 | 4.1 |
| G11 | 800 | 1600 | 607 | 0.8 | 655 | 2.8 | 645 | 0.3 | 660 | 1.9 |
| G12 | 800 | 1600 | 600 | 0.8 | 646 | 5.6 | 633 | 0.3 | 653 | 1.9 |
| G13 | 800 | 1600 | 621 | 0.8 | 671 | 5.4 | 661 | 0.3 | 677 | 1.7 |
| G14 | 800 | 4694 | 3886 | 0.9 | 3987 | 7.3 | 3960 | 0.4 | 3998 | 2.2 |
| G15 | 800 | 4661 | 3855 | 1.0 | 3966 | 6.5 | 3933 | 1.1 | 3968 | 2.1 |
| G16 | 800 | 4672 | 3876 | 0.9 | 3966 | 7.3 | 3934 | 0.4 | 3971 | 2.2 |
| G17 | 800 | 4667 | 3861 | 0.8 | 3956 | 3.9 | 3929 | 1.1 | 3964 | 2.0 |
| G18 | 800 | 4694 | 1080 | 1.0 | 1110 | 7.0 | 1140 | 0.3 | 1188 | 2.2 |
| G19 | 800 | 4661 | 939 | 0.9 | 971 | 7.8 | 1020 | 0.6 | 1062 | 2.3 |
| G20 | 800 | 4672 | 966 | 0.9 | 1004 | 7.7 | 1048 | 0.3 | 1106 | 2.2 |
| G21 | 800 | 4667 | 978 | 0.9 | 1010 | 5.7 | 1063 | 0.4 | 1102 | 2.0 |
| G22 | 2000 | 19990 | 16679 | 1.5 | 17068 | 16.0 | 16964 | 0.8 | 17122 | 4.3 |
| G23 | 2000 | 19990 | 16676 | 1.3 | 17084 | 25.2 | 16954 | 0.8 | 17127 | 4.3 |
| G24 | 2000 | 19990 | 16701 | 1.4 | 17063 | 22.1 | 16959 | 1.0 | 17119 | 4.8 |
| G25 | 2000 | 19990 | 16690 | 1.4 | 17072 | 24.2 | 16971 | 1.0 | 17115 | 4.2 |
| G26 | 2000 | 19990 | 16686 | 1.4 | 17056 | 22.2 | 16941 | 0.9 | 17113 | 4.1 |
| G27 | 2000 | 19990 | 3534 | 1.4 | 3129 | 24.9 | 3814 | 0.8 | 3968 | 4.2 |
| G28 | 2000 | 19990 | 3456 | 1.4 | 3030 | 13.2 | 3752 | 0.8 | 3921 | 4.1 |
| G29 | 2000 | 19990 | 3593 | 2.1 | 3225 | 9.3 | 3899 | 2.4 | 4056 | 4.9 |
| G30 | 2000 | 19990 | 3593 | 1.6 | 3231 | 14.9 | 3887 | 2.1 | 4078 | 5.6 |
| G31 | 2000 | 19990 | 3434 | 1.4 | 3067 | 18.9 | 3773 | 0.8 | 3956 | 5.8 |
| G32 | 2000 | 4000 | 1509 | 1.1 | 1602 | 11.3 | 1591 | 0.3 | 1635 | 2.3 |
| G33 | 2000 | 4000 | 1446 | 1.0 | 1572 | 11.8 | 1553 | 1.1 | 1602 | 2.3 |
| G34 | 2000 | 4000 | 1437 | 1.0 | 1551 | 11.1 | 1546 | 0.4 | 1591 | 2.3 |
| G35 | 2000 | 11778 | 9752 | 1.2 | 9981 | 16.3 | 9897 | 1.9 | 9997 | 3.4 |

Table 7 continued.

| Instance | $|\mathcal{V}|$ | $|\mathcal{E}|$ | ROS | | ANYCSP | | FEM | | PDBO | |
|---|---|---|---|---|---|---|---|---|---|---|
| | | | Obj. ↑ | Time (s) ↓ | Obj. ↑ | Time (s) ↓ | Obj. ↑ | Time (s) ↓ | Obj. ↑ | Time (s) ↓ |
| G36 | 2000 | 11766 | 9744 | 1.1 | 9978 | 18.2 | 9873 | 1.9 | 10003 | 3.5 |
| G37 | 2000 | 11785 | 9766 | 1.1 | 9989 | 17.0 | 9902 | 1.8 | 10004 | 3.1 |
| G38 | 2000 | 11779 | 9795 | 1.2 | 9986 | 4.8 | 9893 | 1.9 | 10000 | 3.1 |
| G39 | 2000 | 11778 | 2533 | 1.4 | 2525 | 9.7 | 2722 | 0.5 | 2857 | 3.8 |
| G40 | 2000 | 11766 | 2461 | 1.3 | 2536 | 16.3 | 2671 | 1.9 | 2833 | 3.3 |
| G41 | 2000 | 11785 | 2524 | 1.2 | 2546 | 17.8 | 2679 | 0.5 | 2834 | 3.8 |
| G42 | 2000 | 11779 | 2582 | 1.2 | 2631 | 18.0 | 2777 | 1.9 | 2926 | 3.4 |
| G43 | 1000 | 9990 | 8312 | 1.1 | 8525 | 4.3 | 8478 | 0.6 | 8566 | 3.3 |
| G44 | 1000 | 9990 | 8349 | 1.4 | 8523 | 12.8 | 8486 | 0.6 | 8550 | 3.1 |
| G45 | 1000 | 9990 | 8354 | 1.6 | 8512 | 11.9 | 8486 | 0.5 | 8559 | 3.0 |
| G46 | 1000 | 9990 | 8346 | 4.1 | 8508 | 7.2 | 8473 | 1.5 | 8558 | 3.0 |
| G47 | 1000 | 9990 | 8338 | 2.7 | 8523 | 10.7 | 8483 | 0.5 | 8569 | 3.1 |
| G48 | 3000 | 6000 | 5920 | 2.7 | 6000 | 7.7 | 6000 | 0.4 | 6000 | 2.3 |
| G49 | 3000 | 6000 | 5914 | 2.3 | 6000 | 6.9 | 6000 | 0.4 | 6000 | 2.4 |
| G50 | 3000 | 6000 | 5924 | 1.9 | 5998 | 9.7 | 5998 | 0.5 | 6000 | 2.7 |
| G51 | 1000 | 5909 | 4897 | 1.0 | 5004 | 8.3 | 4957 | 0.4 | 5012 | 2.3 |
| G52 | 1000 | 5916 | 4910 | 1.0 | 5019 | 9.1 | 4970 | 0.4 | 5012 | 2.3 |
| G53 | 1000 | 5914 | 4918 | 1.0 | 5007 | 4.7 | 4964 | 0.4 | 5015 | 2.2 |
| G54 | 1000 | 5916 | 4886 | 0.9 | 5015 | 8.7 | 4966 | 0.4 | 5013 | 2.2 |
| G55 | 5000 | 12498 | 11992 | 1.4 | 12351 | 27.3 | 12210 | 2.2 | 12345 | 4.3 |
| G56 | 5000 | 12498 | 4126 | 1.7 | 4495 | 31.3 | 4425 | 2.2 | 4663 | 4.0 |
| G57 | 5000 | 10000 | 3688 | 1.5 | 3940 | 29.4 | 3900 | 0.4 | 4028 | 3.6 |
| G58 | 5000 | 29570 | 24482 | 4.2 | 25027 | 46.5 | 24807 | 3.7 | 25088 | 5.4 |
| G59 | 5000 | 29570 | 6549 | 5.2 | 6316 | 43.3 | 6785 | 3.9 | 7190 | 8.0 |
| G60 | 7000 | 17148 | 16496 | 1.9 | 16978 | 41.6 | 16797 | 2.9 | 16961 | 4.5 |
| G61 | 7000 | 17148 | 5989 | 2.0 | 6464 | 41.2 | 6401 | 2.9 | 6730 | 4.8 |
| G62 | 7000 | 14000 | 5149 | 1.9 | 5487 | 37.0 | 5441 | 2.5 | 5635 | 4.3 |
| G63 | 7000 | 41459 | 34331 | 3.0 | 35087 | 49.3 | 34776 | 5.3 | 35194 | 8.0 |
| G64 | 7000 | 41459 | 9399 | 2.8 | 9219 | 48.1 | 9767 | 5.1 | 10333 | 8.4 |
| G65 | 8000 | 16000 | 5914 | 2.2 | 6255 | 44.3 | 6207 | 2.9 | 6425 | 4.5 |
| G66 | 9000 | 18000 | 6739 | 2.5 | 7147 | 39.4 | 7103 | 3.2 | 7338 | 4.5 |
| G67 | 10000 | 20000 | 7364 | 3.0 | 7797 | 55.9 | 7748 | 3.3 | 8015 | 6.0 |
| G70 | 10000 | 9999 | 9983 | 1.9 | 9909 | 16.2 | 9999 | 1.4 | 9999 | 3.3 |
| G72 | 10000 | 20000 | 7435 | 2.9 | 7906 | 58.9 | 7835 | 0.7 | 8111 | 4.7 |
| G77 | 14000 | 28000 | 10559 | 5.3 | 11158 | 84.0 | 11102 | 4.4 | 11467 | 5.0 |
| G81 | 20000 | 40000 | 14907 | 9.7 | 15727 | 115.0 | 15683 | 6.1 | 16191 | 7.9 |

## D.2  EXPERIMENTS FOR GPU EFFICIENCY

In this section we conduct comparison between the CPU/GPU versions of PDBO.

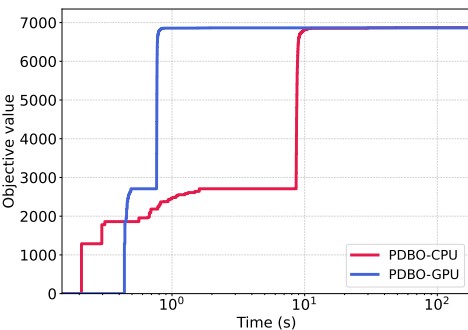

Figure 3: The objective value of Max-Cut on G67, as a function of runtime

Figure 3 compares the solution quality of each method throughout the optimization process, plotting the objective value against runtime under a 180-second time limit, on the G67 instance for Max-Cut. Clearly, `PDBO-GPU` (blue) performs much faster than `PDBO-CPU` (red). The initial latency of `PDBO-GPU` could be attributed to the *just-in-time* nature of Jax.

## D.3  EXPERIMENTS ON MAP

In this section we conduct additional experiments on the Maximum a Posterior (MAP) inference in discrete Markov random fields.

### D.3.1  PROTEIN PREDICTION

In particular, we adopt the protein-prediction dataset (Jaimovich et al., 2006), which is widely adopted in the related literature and competitions (Kappes et al., 2015; Elidan & Globerson, 2011). The dataset includes 8 instances and Table 8 presents the average objective value and running time for each method. Please refer to Kappes et al. (2015) for detailed description on the baselines as well as the formal problem definition.

Table 8: Results for MAP on the protein-prediction dataset

|  | Obj.↓ | Time↓ |
|---|---|---|
| ogm-LBP-LF2 | 52942.95 | 69.86 |
| ogm-LF-3 | 57944.06 | 25.99 |
| ogm-LBP-0.5 | 53798.89 | 60.97 |
| ogm-TRBP-0.5 | 61386.17 | 86.03 |
| ADDD | 106216.86 | 10.97 |
| MPLP | 101531.75 | 86.6 |
| ogm-LP-LP | 102918.41 | 180.72 |
| ogm-ILP | 60047.02 | 2262.83 |
| BRAOBB-1 | 61079.07 | 3600.16 |
| PDBO | **52842.61** | 15.07 |

The results show that `PDBO` is both efficient and effective for handling the MAP problems.

### D.3.2  UAI2022 COMPETITION

We also evaluate PDBO on the datasets from the UAI 2022 competition (`https://uaicompetition.github.io/uci-2022/`), a standard modern benchmark. In particular,

we consider the **Grids** dataset (7 instances in total) that aligns with the binary optimization formulation (1). We compare against three state-of-the-art baselines—`LBP`, `TRBP`, and `Toulbar2`—with their results taken directly from the recent and comprehensive study of Wang et al. (2024):

- Loopy Belief Propagation (denoted as `LBP`) & Tree-reweighted Belief Propagation (denoted as `TRBP`) (Wainwright et al., 2005): Classic and widely-used message-passing algorithms.

- `Toulbar2` (De Givry, 2023): the winner on all MPE and MMAP task categories of UAI 2022 Inference Competition.

Table 9: Results on the UAI 2022 benchmarks, lower values imply better performance

|          | Grids_19 | Grids_21 | Grids_24 | Grids_25 | Grids_26 | Grids_27 | Grids_30 |
|----------|----------|----------|----------|----------|----------|----------|----------|
| LBP      | -2250.440 | -13119.300 | -13210.400 | -2170.890 | -2063.350 | -9024.640 | -2142.890 |
| TRBP     | -2103.610 | -12523.300 | -13260.900 | -2171.050 | -1903.910 | -9019.470 | -2154.910 |
| Toulbar2 | -2643.107 | -18895.393 | -18274.302 | -2620.268 | -3010.719 | -12284.284 | -2984.248 |
| PDBO     | **-2696.341** | **-19235.873** | **-18691.510** | **-2653.470** | **-3021.883** | **-12481.272** | **-2989.997** |

The results are shown in Table 9. `PDBO` achieves the best performance on all seven instances. Notably, it finds these superior solutions within 5.9 seconds on average, while the baselines from Wang et al. (2024) (including `Toulbar2` with a 1,200-second limit) could not match this performance. These results, conducted on a standard modern benchmark and compared against baseline results, robustly demonstrate PDBO's effectiveness and efficiency.

### D.4 COMPARISON AGAINST THE CONTINUATION METHOD

Note that several of our baselines (e.g. `CRA`, `FEM`) employ a continuation method (or annealing strategy) that consider the extension $f(x) + \mu \sum_{i=1}^{n} g(x_i)$, and gradually decreases $\mu$ to simulate the annealing process. While this bears an inherent analogy to `PDBO` with a single-constraint reformulation

$$\min_{x \in [0,1]^n} \quad f(x)$$

$$\text{s.t.} \quad \sum_{i=1}^{n} g(x_i) = 0,$$

we highlight two critical differences:

- **Principled update rule**: The annealing strategy relies on a pre-defined, often heuristic, schedule for descending $\mu$. In contrast, PDBO is derived from a primal-dual perspective, with its update rule for the dual variable explicitly given by GDA.

- **Tighter relaxation**: Let $f_\mu(x) := f(x) + \mu \sum_{i=1}^{n} g(x_i)$. The extension space explored by PDBO, $\mathcal{S} := \{f_y : y \in \mathbb{R}^n\}$, is much larger than the one-dimensional path $\mathcal{S}' := \{f_\mu : \mu \in \mathbb{R}\}$ used by the continuation method. Since optimizing the dual variables can be viewed as constructing an improved extension, PDBO has a greater capacity to find a tighter relaxation. This advantage is concretely illustrated in the QUBO case, where $f(x) = x^\top Q x + q^\top x$ and $g(x) = x^2 - x$. Consider the convex relaxations $\mathcal{S}_{\text{convex}} = \{f_y : y \in \mathbb{R}^n, Q + \text{diag}(y) \succeq 0\}$. As shown in Lemaréchal & Oustry (1999), the tightest relaxation in $\mathcal{S}_{\text{convex}}$ achieves a dual bound equivalent to the SDP relaxation. However, the tightest relaxation in the restricted space $\mathcal{S}'_{\text{convex}} = \{f_\mu : \mu \in \mathbb{R}, Q + \mu I_n \succeq 0\}$ is given by the smallest eigenvalue $\mu = \lambda_{\min}(Q)$, which yields a looser bound.

To empirically validate this advantage, we compare PDBO against a variant using only a single aggregate constraint (denoted as "`PDBO (single)`"). The results confirm that the full primal-dual framework (PDBO) improves over continuation-style baselines (`CRA`, `FEM`), while the additional flexibility of $n$ constraints (vs. one) yields further performance gains.

Table 10: Computational results for Max-Cut

| Method | G67 (n=10k) | | G70 (n=10k) | | G72 (n=10k) | | G77 (n=14k) | | G81 (n=20k) | |
|---|---|---|---|---|---|---|---|---|---|---|
| | Obj ↑ | Time ↓ | Obj ↑ | Time ↓ | Obj ↑ | Time ↓ | Obj ↑ | Time ↓ | Obj ↑ | Time ↓ |
| CRA | 5948 | 53.7 | 9240 | 51.7 | 6058 | 53.9 | 8720 | 75.9 | 12450 | 120.4 |
| FEM | 6782 | 2.4 | 5120 | 0.2 | 6824 | 2.6 | 9688 | 4.0 | 13684 | 7.5 |
| PDBO(single) | 6844 | 2.0 | 9509 | 1.1 | 6880 | 1.4 | 9782 | 1.8 | 13786 | 2.5 |
| PDBO | **6872** | 2.5 | **9537** | 1.9 | **6906** | 2.0 | **9812** | 2.1 | **13852** | 3.3 |

### D.5 DUAL CERTIFICATE

When the initial value $y^0$ is sufficiently large (e.g. $\bar{y} \geq -\lambda_{min}(W)$ for max-cut), the Lagrangian extension $f_y(x) := L(x, y)$ turns out to be convex in $x$. In this regime, we are able to globally minimize $f_y(x)$ via classic gradient descent (albeit PDBO employs a single gradient descent step instead), the corresponding optimal value would provide a valid dual bound on the optimal objective value, serving as a dual certificate. We present the corresponding dual bounds of the Max-Cut problem implied by PDBO in Table 11.

Table 11: Dual bounds for Max-Cut, implied by PDBO

| Max-Cut | G67 | G70 | G72 | G77 | G81 |
|---|---|---|---|---|---|
| primal bound | 6872 | 9537 | 6906 | 9812 | 13852 |
| dual bound | 8843.7 | 13891.5 | 8906.1 | 12767.7 | 18172.2 |

