# OpenReview forum: "Smoothing Binary Optimization: A Primal-Dual Perspective"
_ICLR.cc/2026/Conference — Submitted to ICLR 2026_

### Official Review · Reviewer_s1Wx · 2025-10-30

**Soundness:** 3
**Presentation:** 4
**Contribution:** 3
**Rating:** 4
**Confidence:** 4

**Summary:**

The authors propose a heuristic method to approximately solve a wide class of combinatorial optimization problems, which seek to minimize a real-valued function $F(x)$ over a binary vector $x\in\\{0,1\\}^n$. If such a function is replaced with its multilinear extension $f(x)$, the relaxation in which we minimize $f(x)$ over $x\in[0,1]^n$ is exact, i.e., equal to the original problem. Multilinear extensions are compact for many common combinatorial problems. This relaxation is of course non-convex.

To this relaxation, the authors add constraints to enforce integral solutions. These have the form $g(x_i)=0$ where $g$ is a convex function on $[0,1]$ that vanishes at and only at 0 and 1, such as $g(x_i)=x_i^2-x_i$. Using Lagrangian $L(x,y)=f(x)+\\sum_i y_i g(x_i)$, these constraints are then dualized, where the duality gap is shown to be zero.

Now gradient descent-ascent method is used to find a saddle point of the Lagrangian. This algorithm behaves much like a continuation method: it starts with a (large) positive dual variables $y$, in which case the function $x\\mapsto L(x,y)$ is convex, so global approximate solutions are preferred. During GDA, the components of $y$ are slowly decreasing, which is making the function non-convex and becoming more and more greedy. The algorithm is augmented with a heuristic that pushes $x_i$ away from $1/2$, to avoid possible trivial saddle points with $x_i=1/2$. The algorithm is very simple.

The convergence analysis shows that $y_i$ are bounded below, $x$ converge to integral solution, and convergence rate is linear.

In experiments, the authors compare the method to several SOTA heuristics for approximately solving large-scale combinatorial optimization problems. The problems used are max-cut, max-k-cut, max independent set, and max-sat. The instances are taken from public datasets, which are mostly random graphs. The reported results show that the method is always competitive and often the best in objective value, and one of the fastest.

**Strengths:**

The approach is very simple and elegant, resulting in a very simple (a few lines) but versatile algorithm. The theoretical analysis is elegant and also relatively simple. Each experiment is done carefully (but see below).

**Weaknesses:**

I have two major comments.

FIRST comment: The experimental comparison is too limited (the choice of problems, of instances, and also of competing methods). While one can always say this, in this case it is justified. Let me argue why. On line 102, the authors write that the method provably converges to a near-optimal solution. But this is not true: the presented theory says absolutely nothing about the quality of found suboptima. Thus, the method is only a heuristic (however elegant). Thus, justifying its practical utility fully relies on extensive experimental comparison.

The choice of tested problems is limited. For example, I am missing the MAP inference problem in MRF (aka discrete energy minimization), which has been subject to very intensive research in computer vision and machine learning, see e.g. experimental studies [1], [2] (with public datasets). Another possible problem is QAP, which is known to be very challenging; public QAP datasets are e.g. [4], [5].

The choice of instances (for tested problems) is limited to mostly random instances. But it is known that random instances tend to be much easier to optimize that real-world ones.

The choice of competing methods is also limited, biased to recent works. Of course, for MAP inference and QAP one should use SOTA dedicated methods, as in [1, 2, 3]. But even for the tested problems, off-the-shelf global ILP solvers (like Gurobi) might be worth trying. I am not entirely sure that they could handle the (large) tested instances, but it was observed (eg, in computer vision) that they can be surprisingly fast on many large ILP instances from practice. Perhaps, this would not be very competitive time-wise, but it would give  global optima to compare to.

SECOND comment: I am not sure about the level of novelty. Many heuristics have been shown to perform surprisingly well on certain classes of large-scale combinatorial problem instances. The presented method is similar to several such methods. One example is [3], which solves the multilinear formulation of the MAP inference problem approximately by ADMM (though on a different problem reformulation), with great results. This is probably subsumed by many later works on applying ADMM directly to nonconvex/integer problems.

The presented method can be easily reformulated by doing (block-)coordinate descent on primal variables, instead of gradient descent. By augmenting the Lagrangian, one gets ADMM.

Moreover, the method is very similar to continuation methods. The continuation method could minimize $f(x)+\\mu \\sum_i g_i(x_i)$, starting with a large positive $\\mu$ (which would make the function convex) and then slowly decreasing $\\mu$. In fact, this is almost what is going on in GDA, up to the fact that instead of a single variable $\\mu$ we have a vector $y$, whose components are decreased automatically as a result of gradient ascent. I tend to believe that such a continuation method would yielded similar experimental results. This would make the paper significantly less novel, from the theory side.

Note, instead of $n$ constraints $g(x_i)=0$ one could use a single constraint $\sum_i g(x_i)=0$, which would enforce iintegrality of $x$ in the same way. For $g(x_i)=x_i^2-x_i$, this would read $(x-1)^T x=0$. In this case, there would be only one $y$-variable, like in continuation method. When using GDA, would this give similar experimental results?

It would be useful to discuss the relation to other similar methods in more detail in the paper (or supplement). And try to experimentally identify the features of the method that make it perform well (a kind of ablation study).

Minor remarks:

- In Def 2, the term 'multilinear' collides with its standard meaning in linear algebra.

- In section 2, it is not clear if Lemma 1 and Prop 1 are novel or known before. Please make this clear.

- In Algorithm 1, the projection on a non-convex set (the union of intervals) does not exist, strictly formally. If $x=1/2$, there are two nearest points.

- A more detailed discussion around the example on line 222 might be useful. In fact, it means that GDA does not find a stationary point in this case. If GDA always found a stationary point of $L(x,y)$, the $\\delta$-heuristic would not be needed. In this respect, a nonconvex version of ADMM with convergence guarantees might be a better choice than GDA, cf. [6].

[1] Szeliski et al: A Comparative Study of Energy Minimization Methods for Markov Random Fields, ECCV 2006

[2] Kappes et al: A Comparative Study of Modern Inference Techniques for Structured Discrete Energy Minimization Problems. IJCV 2015

[3] Le-Huu, Paragios: Continuous Relaxation of MAP Inference: A Nonconvex Perspective. CVPR 2018

[4] Haller et al: A comparative study of graph matching algorithms in computer vision. ECCV 2022

[5] Burkard et al: QAPLIB -- a quadratic assignment problem library. J. Global Optim. 10(4), 391–403 (1997)

[6] Wang et al: Global Convergence of ADMM in Nonconvex Nonsmooth Optimization. 2018

**Questions:**

1) Is there an obstacle for testing on other combinatorial problems in experiments, such as MAP inference or QAP, and real-world instances, such as in [2]? Are there arguments to believe that the method would be competitive also on these problems?

2) Is there an argument why the method is qualitatively different from well-known continuation methods (as I described above), augmented with the $\\delta$-heuristic? Is there an argument why such a continuation method would perform much worse?

3) Please discuss novelty of your approach in more detail, comparing to similar methods and emphasizing the features that are essential for success.

---

> ### Author Response · Authors · 2025-11-21
> **Rebuttal Part I**
>
> >### **W1 FISRT comment**
> > **W1.1**: The experimental comparison ...fully relies on extensive experimental comparison.
>
> Thank you for this feedback. We agree that PDBO is best characterized as an **elegant, scalable, and theory-grounded heuristic** for unconstrained binary optimization. Consequently, its practical utility must be justified empirically. To this end, we conduct extensive experiments across **five distinct, widely used problem classes** (Max-Cut, MIS, Max-k-SAT, Max-k-Cut, and **MAP**). The results consistently demonstrate the method's **effectiveness** in finding high-quality solutions and its **efficiency**, particularly on large-scale problems, thereby solidifying its practical value.
>
>
> > **W1.2** The choice of tested problems is limited. ... But it is known that random instances tend to be much easier to optimize that real-world ones.
>
> Thank you for this valuable feedback regarding our experimental benchmarks.
> - **Current Benchmarks**: Our selected datasets—Gset for Max-Cut and dense RRGs for MIS—are **computationally challenging and represent widely-adopted** standards in the literature [7-11]. We note that the dense RRGs (degree 100) used in our study are significantly more difficult than the sparser graphs (e.g., degree 3) often used in prior work, as detailed in Section 4.3.2.
> - **QAP**: We agree that QAP is a valuable benchmark. However, PDBO in its current form is **not directly applicable**, as QAP involves **assignment constraints** on the decision variables, which do not fit the unconstrained binary formulation of our problem (1).
> - **MAP**: We thank the reviewer for this suggestion. To align with our binary variable formulation, we have now included experiments on a **protein prediction** task, a **well-established binary MAP** benchmark [2, 12]. This addition demonstrates our method's applicability to a real-world, structured problem from this domain.
>
>
> > **W1.3** The choice of competing methods ....
>
>
> Thanks for your suggestion. We now consider more basline algorithms
> - **SOTA dedicated methods**: For the MAP problem, we compare against multiple specialized SOTA baselines including ADDD[15], MPLP[16], and BRAOBB[17].
> - **Off-the-shelf global ILP solvers**: We now include Gurobi as suggested.
>
>
>
> ### **1. Comparison against Gurobi**
>
> We also conduct additional experiments on Max-Cut, Max-k-Cut, Max-k-SAT and MIS with Gurobi 10.0.2. The results are presented in the following table.
> - The results demonstrate that PDBO achieves significantly better performance on Max-3-Cut, MIS, and MaxSAT instances.
> - For Max-Cut, PDBO finds superior solutions for instances G70 and G81, while Gurobi performs better on G67, G72, and G77, albeit at a **substantially higher computational cost**. It is worth noting, **as highlighted in [14]**, that **recent versions of Gurobi have seen major improvements on QUBO and Max-Cut problems**, likely due to the integration of **specialized heuristics**.
> - We depict Gurobi's objective values against runtime in **Figure 2**, from which we conclude that PDBO **achieves substantially better solutions than Gurobi for the majority of the runtime**.
>
> These new comparisons with Gurobi have been added to the revised manuscript.
>
> | Max-Cut | G67-obj $\uparrow$|G67-time| G70-obj$\uparrow$|G70-time | G72-obj$\uparrow$|G72-time|G77-obj$\uparrow$|G77-time|G81-obj$\uparrow$|G81-time|
> |-|-|-|-|-|-|-|-|-|-|-|
> PDBO| 6872 | 2.5 | **9537** | 1.9 | 6906 | 2.0 | 9812 | 2.1 | **13852** | 3.3
> Gurobi|**6944** | 151.9 | 9514 | 133.3 | **6990** | 171.6 | **9882** | 175.1 | 13848 | 179.8
>
> | Max-3-Cut | G67-obj $\uparrow$|G67-time| G70-obj$\uparrow$|G70-time | G72-obj$\uparrow$|G72-time|G77-obj$\uparrow$|G77-time|G81-obj$\uparrow$|G81-time|
> |-|-|-|-|-|-|-|-|-|-|-|
> PDBO | **8015** | 6.0 | **9999** | 3.3 | **8111** | 4.7 | **11467** | 5.0 | **16191** | 7.9 |
> Gurobi| 7855 | 178.1 | **9999** | 0.7 | 8045 | 174.6 | 11101 | 179.9 | 15146 | 179.9 |
>
>
>
> | MIS | ($10^4$, 3)-obj$\uparrow$| ($10^4$, 3)-time |($10^4$, 100)-obj|($10^4$, 100)-time|($5\times10^4$, 3)-obj$\uparrow$|($5\times10^4$, 3)-time|($5\times 10^4$, 100)-obj$\uparrow$|($5\times 10^4$, 100)-time|
> |-|-|-|-|-|-|-|-|-|
> PDBO|$\bf{4431.9_{\pm  3.1 }}$ |$0.6 _{\pm  0.0 }$ |$\bf{603.0 _{\pm  3.3 }}$ |$3.6 _{\pm  0.3 }$ |$\bf{22127.3 _{\pm  10.3 }}$ |$1.1 _{\pm  0.1 }$ |$\bf{3006.2 _{\pm  6.2 }}$ |$14.6 _{\pm  0.6 }$ |
> Gurobi |$4141.8 _{\pm  19.4 }$ |$147.2 _{\pm 24.8 }$ |$448.0 _{\pm  5.1 }$ |$0.1 _{\pm  0.0 }$ |$20144.3 _{\pm  181.7 }$ |$96.6 _{\pm  49.9 }$ |$2235.6 _{\pm  12.7 }$ |$0.1 _{\pm  0.0 }$ |
>
>
> | MaxSAT |3CNF-obj$\downarrow$|3CNF-time| 4CNF-obj$\downarrow$|4CNF-time | 5CNF-obj$\downarrow$|5CNF-time|
> |-|-|-|-|-|-|-|
> PDBO |$\bf{1583.0 _{\pm  15.7 }}$ |$3.5 _{\pm  1.6 }$ |$\bf{1230.2 _{\pm  14.4 }}$ |$4.3 _{\pm  1.8 }$ |$\bf{1148.1 _{\pm  11.8 }}$ |$9.3 _{\pm  3.1 }$ |
> Gurobi |$9322.6 _{\pm  64.2 }$ |$0.0 _{\pm  0.1 }$ |$9329.2 _{\pm  69.7 }$ |$0.0 _{\pm  0.0 }$ |$9310.7 _{\pm  75.4 }$ |$0.1 _{\pm  0.0 }$ |

---

> ### Author Response · Authors · 2025-11-21
> **Rebuttal Part II**
>
> ### **2. Additional Experiments on MAP**
>
> Following the reviewer's suggestion, we have conducted additional experiments on the **protein-prediction** dataset, a standard benchmark for **MAP inference** [2]. This dataset consists of 8 challenging instances, each containing over **14,000 variables**. We evaluated PDBO on these instances with a time limit of 180 seconds each and compared its performance against the **published baseline results from [2]**. The averaged results across all instances are summarized below (a lower objective value indicates better performance):
>
> |Protein-prediction|Obj $\downarrow$|Time|
> |-|-|-|
> |ogm-LBP-LF2 [18, 19, 20]|52942.95|69.86|
> ogm-LF-3 [18, 20]|57944.06|25.99|
> ogm-LBP-0.5 [18, 19]|53798.89|60.97|
> ogm-TRBP-0.5 [18, 21]|61386.17|86.03|
> ADDD [15]|106216.86|10.97|
> MPLP [16]|101531.75|86.6|
> ogm-LP-LP [18]|102918.41|180.72|
> ogm-ILP [18]|60047.02|2262.83|
> BRAOBB-1 [17]|61079.07|3600.16|
> PDBO|**52842.61**|15.07
>
> The results demonstrate the superiority of PDBO, as it achieves a better (lower) objective value and/or superior computational efficiency compared to the advanced MAP algorithms benchmarked in [2].
>
> We append this part in our revised manuscript in **Appendix D.3**.
>
> > ### **W2. SECOND comment**
> > **W2.1**. I am not sure ... By augmenting the Lagrangian, one gets ADMM.
>
> Thanks for raising this. We clarify the difference between PDBO and the ADMM in [3].
> - **Different frameworks**: ADMM is derived via variable splitting, while PDBO dualize the binary constraint $g(x_i)=0$ and follows the elegant Lagrangian dual framework.
> - **Exclude augmented penalties**: In PDBO, the Lagragian term $\sum\limits_{i=1}^n y_ig(x_i)$ itself can be considered a penalty for fractionality, with coefficient $-y_i$ increasing. (As mentioned in our Section 3.2.1).
> - **Principled update**: The penalty coefficient $-y_i$ in PDBO is dynamically adjusted following GDA. However, in ADMM one needs to carefully decide the penalty coefficient $\rho$ and design its update rule.
> - **Convergence to Integer**: PDBO guarantees convergence to a binary point, while ADMM relies on external rounding strategies (e.g. BCD [3]).
>
>
> > **W2.2** Moreover, the method is very similar to continuation methods. ..
>
> We thank the reviewer for this insightful observation, we will clarify the **difference between PDBO and the continuation methods**.
> - **Principled update**: Existing baselines like CRA and FEM are inherently continuation methods that rely on a predefined scheduling rule (e.g., exponential or inverse-proportional decay [7]) to manually decrease the penalty parameter $\mu$. In contrast, our primal-dual framework automatically adjusts the dual variables $y$ in a theoretically grounded manner, enabling adaptive, instance-specific penalty modulation rather than relying on a fixed heuristic schedule.
> - **Larger extension space**: Denote $f_y(x)=f(x)+\sum\limits_{i=1}^ny_ig(x_i)$ and $f_\mu(x)=f(x)+\mu\sum\limits_{i=1}^ng(x_i)$. Clearly both functions serve as an extensions of $F$ for any $y\in\mathbb R^n$ and $\mu\in\mathbb R$. The extension space explored by PDBO, $\mathcal{S} = \{f_y : y \in \mathbb{R}^n\}$, is much larger than the one-dimensional path $\mathcal{S}' = \{f_\mu : \mu \in \mathbb{R}\}$ used by the continuation method. Since optimizing the dual variables can be viewed as constructing an improved extension, PDBO has a greater capacity to find a tighter relaxation.
> - **Tighter (convex) relaxation**: Consider QUBO, i.e. $f(x)=x^\top Qx+q^\top x$ with $g(x_i)=x_i^2-x_i$, and focus on the convex relaxations $\mathcal S_{convex}=\\{f_y:y\in\mathbb R^n,Q+diag(y)\succeq 0\\}$.
> As shown in [13], the tightest relaxation in $\mathcal S_{convex}$ achieves a dual bound equivalent to the SDP relaxation. However, the tightest relaxation in the restricted space $\mathcal S_{convex}' =\\{f_\mu : \mu\in \mathbb R, Q + \mu I_n \succeq 0\\}$ is given by the smallest eigenvalue $\mu = \lambda_{\min}(Q)$, which yields a looser bound.
>
> To empirically validate this advantage, we **compare PDBO against a variant using only a single aggregate** constraint ("PDBO (single)"). The results confirm that **the full primal-dual framework (PDBO) improves over continuation-style baselines (CRA, FEM), while the additional flexibility of $n$ constraints (vs. one) yields further performance gains.**
>
> |Max-cut|G67-obj|G67-time|G70-obj|G70-time|G72-obj|G72-time|G77-obj|G77-time|G81-obj|G81-time|
> |-|-|-|-|-|-|-|-|-|-|-|
> |CRA|5948|53.7|9240|51.7|6058|53.9|8720|75.9|12450|120.4|
> |FEM|6782|2.4|5120|0.2|6824|2.6|9688|4.0|13684|7.5|
> |PDBO (single) |6844|2.0|9509|1.1|6880|1.4|9782|1.8|13786|2.5
> |PDBO|**6872**|2.5|**9537**|1.9|**6906**|2.0|**9812**|2.1|**13852**|3.3|

---

> > ### Author Response · Authors · 2025-11-21
> > **Rebuttal Part III**
> >
> > > **W2.3** It would be useful..
> >
> > We now provide detailed discussion on the relation to other similar methods in our revised manuscript in **Appendix D.4**.
> >
> > ### Minor remarks:
> >
> > > **R1** In Def 2 ..
> >
> > Thanks for the pointing this out, we agree with the reviewers that the definition should be "A function $f:\mathbb{R}^n \rightarrow \mathbb{R}$ is called multilinear if it is linear in each variable when all other variables are held fixed." We now correct it in the revised manuscript.
> >
> > > **R2** In section 2..
> >
> > Thanks for the suggestion, these properties are presented in Preliminaries and are therefore known results. As also suggested by reviewer DrfU, we will give a brief introduction at the beginning of Section 2 to clarify this in our revised manuscript.
> >
> > > **R3** In Algorithm 1...
> >
> > Thanks for catching this, we correct it as $$x_i^{t+1} \gets \begin{cases}
> >             \frac{1}{2}-\delta & \text{if } x_i^t \leq \frac{1}{2}, \\\\
> >             \frac{1}{2}+\delta & \text{otherwise.}
> >             \\end{cases}$$
> > and we update this in our revised manuscript.
> >
> > > **R4** A more detailed ..
> >
> > Thanks for your suggestion.
> > - We clarify that instead of stationary points, PDBO aims at binary points, and the GDA procedure will stablize due to projection of $x$ onto $[0,1]^n$.
> > - The $\delta$-heuristic is to avoid the fractional stationary point.
> > - By setting $g(x)$ to be symmetric on $[0,1]$, the only stationary point of PDBO is $(\frac12,...,\frac12)$. Therefore, to avoid this fractional stationary point, the $\delta-$perturbation is employed.
> >
> >
> > ### Questions:
> >
> > > **Q1**. Is there..?
> >
> > Please refer to our reply to **W1**.
> >
> > > **Q2**. Is there ..?
> >
> > Please refer to our reply to **W2**.
> >
> > > **Q3**. Please ..
> >
> > Please refer to our reply to **W2**.
> >
> >
> > **References**
> >
> > [1] Szeliski et al: A Comparative Study of Energy Minimization Methods for Markov Random Fields, ECCV 2006
> >
> > [2] Kappes et al: A Comparative Study of Modern Inference Techniques for Structured Discrete Energy Minimization Problems. IJCV 2015
> >
> > [3] Le-Huu, Paragios: Continuous Relaxation of MAP Inference: A Nonconvex Perspective. CVPR 2018
> >
> > [4] Haller et al: A comparative study of graph matching algorithms in computer vision. ECCV 2022
> >
> > [5] Burkard et al: QAPLIB -- a quadratic assignment problem library. J. Global Optim. 10(4), 391–403 (1997)
> >
> > [6] Wang et al: Global Convergence of ADMM in Nonconvex Nonsmooth Optimization. 2018
> >
> > [7] Shen, Zi-Song, et al. "Free-energy machine for combinatorial optimization." Nature Computational Science (2025): 1-11.
> >
> > [8] Ichikawa, Yuma. "Controlling continuous relaxation for combinatorial optimization." Advances in Neural Information Processing Systems 37 (2024): 47189-47216.
> >
> > [9] Qiu, Yeqing, et al. "ROS: A GNN-based Relax-Optimize-and-Sample Framework for Max-k-Cut Problems." Forty-second International Conference on Machine Learning.
> >
> > [10] Chen, C., Chen, R., Li, T. et al. A Monte Carlo Policy Gradient Method with Local Search for Binary Optimization. Math. Program. (2025).
> >
> > [11] Schuetz, Martin JA, J. Kyle Brubaker, and Helmut G. Katzgraber. "Combinatorial optimization with physics-inspired graph neural networks." Nature Machine Intelligence 4.4 (2022): 367-377.
> >
> > [12] G. Elidan and A. Globerson. The probabilistic inference challenge (PIC2011). http://www.cs.huji.ac.il/project/PASCAL/.
> >
> > [13] Lemaréchal, Claude, and François Oustry. Semidefinite relaxations and Lagrangian duality with application to combinatorial optimization. Diss. INRIA, 1999.
> >
> > [14] Daniel Rehfeldt, Thorsten Koch, and Yuji Shinano. Faster exact solution of sparse maxcut and qubo problems. Mathematical Programming Computation, 15(3):445–470, 2023.
> >
> > [15] Martins, André FT, et al. "An augmented Lagrangian approach to constrained MAP inference." ICML. Vol. 2. 2011.
> >
> > [16]Globerson, Amir, and Tommi Jaakkola. "Fixing max-product: Convergent message passing algorithms for MAP LP-relaxations." Advances in neural information processing systems 20 (2007).
> >
> > [17] Otten, Lars, and Rina Dechter. "Anytime AND/OR depth-first search for combinatorial optimization." Ai Communications 25.3 (2012): 211-227.
> >
> > [18] Andres, Bjoern, Thorsten Beier, and Jörg H. Kappes. "OpenGM: A C++ library for discrete graphical models." arXiv preprint arXiv:1206.0111 (2012).
> >
> > [19] Szeliski, Richard, et al. "A comparative study of energy minimization methods for markov random fields with smoothness-based priors." IEEE transactions on pattern analysis and machine intelligence 30.6 (2008): 1068-1080.
> >
> > [20] Andres, Bjoern, et al. "The lazy flipper: Efficient depth-limited exhaustive search in discrete graphical models." European Conference on Computer Vision. Berlin, Heidelberg: Springer Berlin Heidelberg, 2012.
> >
> > [21] Wainwright, Martin J., Tommi S. Jaakkola, and Alan S. Willsky. "MAP estimation via agreement on trees: message-passing and linear programming." IEEE transactions on information theory 51.11 (2005): 3697-3717.

---

> > > ### Comment · Reviewer_s1Wx · 2025-11-26
> > > **Res[once to Rebuttal**
> > >
> > > Thank you for your extensive reply to my review. I appreciate your explanations and mainly new experiments on MAP. I like your remark on SDP vs. eigenvalue relaxation of QUBO.
> > > However, some issues remain and some newly occurred:
> > >
> > > On line 103 of the revised paper, the claim remains that the method provably converges to a near-optimal solution. Let me repeat that this is false: there is no theoretical result that guarantees any bound on the distance of the found suboptimum to the global optimum. Perhaps, by `provably' you mean 'experimentally' but this is misleading.
> > >
> > > In new experiments with Gurobi (new lines in Tables 1, 2, 3, 4), it is often the case that Gurobi (which is a global solver) finds a worse objective value than PDBO before exhausting the time limit (180 seconds). I understand that the reported value is averages over multiple instances. But in Tables 2 and 3, Gurobi stops after 0.1 sec (much before the limit 180 sec) but fails to find the global optimum. This makes me very suspicious about the experiments.
> > >
> > > In MAP experiments on protein predicion, it is very nice that PDBO performs best of all the methods tested in the empirical study [Kappes et al, 2015]. This surprises me because protein folding instances are not dense, and I thought that primal heurstics (e.g., PDBO) shine mainly on dense instances. To be honest, this also makes me somewhat suspicious.
> > >
> > > It is a pity that PDBO was not tested on more instances in [Kappes et al], showing thus its strenghts and weaknesses much more clearly. Note that OpenGM is a ready testbed for this, it would take only to plug PDBO into it. Now the results of the experiments remain somewhat inconclusive.
> > >
> > > In summary, the rebuttal was useful and clarified some of my concerns (though raised some new ones). But I still believe that the paper should have been written more thoroughly already on submission and that, if accepted, it will be 'half-baked', leaving many questions unanswered. But this is the current publication culture, unfortunately..
> > >
> > > I am willing to raise my evaluation to 5. If the paper is accepted, I kindly ask the authors to include all the material (especially experiments) added to the paper after the rebuttal to the final paper/supplement.

---

> > > > ### Author Response · Authors · 2025-11-28
> > > > **Discussion Part I**
> > > >
> > > > > Thank you for your extensive reply to my review. I appreciate your explanations and mainly new experiments on MAP. I like your remark on SDP vs. eigenvalue relaxation of QUBO. However, some issues remain and some newly occurred:
> > > >
> > > > Thank you for your thoughtful feedback and for acknowledging our additions. We appreciate this opportunity to address the remaining points and have incorporated corresponding revisions into our updated manuscript (**see the updated manuscript**).
> > > >
> > > > > **New W1**: On line 103 of the revised paper, the claim remains that the method provably converges to a near-optimal solution. Let me repeat that this is false: there is no theoretical result that guarantees any bound on the distance of the found suboptimum to the global optimum. Perhaps, by `provably' you mean 'experimentally' but this is misleading.
> > > >
> > > > Thank you for pointing this out. We have corrected the phrasing to state that the method "**converges to binary solutions in sublinear time**" and have updated this in our revised manuscript.
> > > >
> > > >
> > > > > **New W2**: In new experiments with Gurobi (new lines in Tables 1, 2, 3, 4), it is often the case that Gurobi (which is a global solver) finds a worse objective value than PDBO before exhausting the time limit (180 seconds). I understand that the reported value is averages over multiple instances. But in Tables 2 and 3, Gurobi stops after 0.1 sec (much before the limit 180 sec) but fails to find the global optimum. This makes me very suspicious about the experiments.
> > > >
> > > > We thank the reviewer for raising this important point regarding the Gurobi timings.
> > > > - As noted in our manuscript, "**The columns labeled “Time” indicates the earliest time at which the method attains the solution corresponding to the reported objective value.**" Thus, the "Time" column records the earliest moment within the 180-second budget at which the best-reported solution was found. **The short times (e.g., 0.1 seconds) for Gurobi in Tables 2 and 3 indicate that it found its best incumbent solution very quickly but failed to improve it throughout the remainder of the time limit**. For example, on the MIS instance with $(10^4,100)$, Gurobi found a solution with an objective value of 448 (on average) in the first 0.1 seconds but could not find a better solution in the subsequent 179.9 seconds.
> > > > - We emphasize that Gurobi did not exhaust its global search capabilities within the 180-second limit. While it might require substantially more time to prove global optimality, our results show that **under this practical time constraint**, Gurobi's performance was limited to the **suboptimal solution it found almost immediately**. This highlights the challenging nature of these large-scale instances where finding improving solutions becomes increasingly difficult.
> > > >
> > > >
> > > > > **New W3**: In MAP experiments on protein predicion, it is very nice that PDBO performs best of all the methods tested in the empirical study [Kappes et al, 2015]. This surprises me because protein folding instances are not dense, and I thought that primal heurstics (e.g., PDBO) shine mainly on dense instances. To be honest, this also makes me somewhat suspicious.
> > > >
> > > >
> > > > We thank the reviewer for this thoughtful observation. We understand the source of surprise, as dense problems are often where traditional solvers struggle most, creating a nice niche for new heuristics.
> > > > - However, PDBO's strength is not solely tied to density. Its performance stems from its general-purpose, first-order nature, which allows it to **navigate complex objective landscapes efficiently**. The protein-prediction instance, while not dense, possesses a challenging and highly non-convex energy landscape. In such cases, PDBO's strategy of **evolving from convex to nonconvex** helps escape shallow local minima that trap more localized search heuristics.
> > > > - This is consistent with our observations on the **MIS dataset** (Table 2 in the manuscript): PDBO delivers superior solutions on both **dense (100-RRGs, degree=100)** and **sparse (3-RRGs, degree=3)** instances, with the performance gap being more pronounced on the former. This pattern demonstrates that PDBO is a robust and versatile optimizer, effective across a spectrum of problem structures.
> > > >
> > > > We now further clarify this in the revised manuscript.

---

> > > > > ### Author Response · Authors · 2025-11-28
> > > > > **Discussion Part II**
> > > > >
> > > > > > **New W4**: It is a pity that PDBO was not tested on more instances in [Kappes et al], showing thus its strenghts and weaknesses much more clearly. Note that OpenGM is a ready testbed for this, it would take only to plug PDBO into it. Now the results of the experiments remain somewhat inconclusive.
> > > > >
> > > > > We agree that more extensive testing is valuable for a conclusive evaluation. We would like to first clarify the practical constraints with the **[Kappes et al.] benchmark** and then present significant **new results** on a standard competition dataset that directly address this point.
> > > > >
> > > > > **1. Clarification on [Kappes et al.] Benchmark:**
> > > > >
> > > > > The majority of instances in [Kappes et al.] benchmark involve variables with multiple labels (**k > 2**), which **fall outside the scope of PDBO** as it is designed for binary optimization (**k = 2**). Among the few binary instances, several (**dtf-chinese-char** and **cell-tracking**) are unfortunately **no longer accessible** via the provided **OpenGM link** [4] (http://hci.iwr.uni-heidelberg.de/opengm2/). For the dtf-chinese-char dataset, we identified an alternative resource (http://ai.kaist.ac.kr/Resource/dbase/Hanja/HanjaDB2.htm), but it is also no longer active. Therefore, the **protein-prediction dataset remained the only publicly accessible and viable binary benchmark** from this study, forming the basis for our reported comparison.
> > > > >
> > > > > **2. New Extensive Evaluation on UAI 2022 Competition Benchmarks:**
> > > > > To provide a more comprehensive evaluation, we have now tested PDBO on the **Grids dataset (7 instances in total)** from the **UAI2022 inference competition** (https://uaicompetition.github.io/uci-2022/), a standard modern benchmark. We compare against three state-of-the-art baselines--**LBP**, **TRBP**, and **Toulbar2**--with their results taken directly from the recent and comprehensive study of [3]:
> > > > > - Loopy Belief Propagation (denoted as "**LBP**") & Tree-reweighted Belief Propagation (denoted as "**TRBP**") [1]: Classic and widely-used message-passing algorithms, **also considered as baseline algorithms in [Kappes et al.]** (see Section 4.3 of their work) as well as **OpenGM** .
> > > > > - **Toulbar2** [2]: The **winner of the UAI 2022 Competition** on MAP inference tasks (see https://uaicompetition.github.io/uci-2022/results/current-leader-board), representing the current state-of-the-art. As reported in [3], it was run with a 1,200-second time limit.
> > > > >
> > > > > The results, showing the best-found objective values (lower is better), are presented below:
> > > > > ||Grids_19 $\downarrow$|Grids_21$\downarrow$|Grids_24$\downarrow$|Grids_25$\downarrow$|Grids_26$\downarrow$|Grids_27$\downarrow$|Grids_30$\downarrow$|
> > > > > |-|-|-|-|-|-|-|-|
> > > > > LBP|-2250.440|-13119.300|-13210.400|-2170.890|-2063.350|-9024.640|-2142.890|
> > > > > TRBP|-2103.610|-12523.300|-13260.900|-2171.050|-1903.910|-9019.470|-2154.910|
> > > > > Toulbar2|-2643.107|-18895.393|-18274.302|-2620.268|-3010.719|-12284.284|-2984.248|
> > > > > PDBO|**-2696.341**|**-19235.873**|**-18691.510**|**-2653.470**|**-3021.883**|**-12481.272**|**-2989.997**|
> > > > >
> > > > > **PDBO achieves the best performance** on all seven instances. Notably, it finds these superior solutions **within 5.9 seconds on average**, while the baselines from [3] (including Toulbar2 with a 1,200-second limit) could not match this performance. These results, conducted on a standard modern benchmark and compared against cited baseline results, robustly demonstrate PDBO's effectiveness and efficiency. We include this analysis in **Appendix D.3.2** of the revised manuscript.
> > > > >
> > > > >
> > > > >
> > > > > > In summary,  .... include all the material (especially experiments) added to the paper after the rebuttal to the final paper/supplement.
> > > > >
> > > > > We sincerely thank the reviewer for their thoughtful engagement. We are grateful for the constructive feedback, which has been invaluable in strengthening our work. We fully agree with the importance of a thorough presentation. In response to all reviewer comments, we have conducted significant new experiments and added clarifying explanations throughout the rebuttal. We hereby **commit to incorporating all this new material**—including the comprehensive comparisons on the UAI 2022 Grids dataset, and the analysis with Gurobi—into the final version of the paper and the supplementary material, ensuring it is far more complete and leaves fewer questions unanswered.
> > > > >
> > > > >
> > > > > ### **References**
> > > > >
> > > > > [1] M.J. Wainwright, T.S. Jaakkola, and A.S. Willsky. Map estimation via agreement on trees: message-passing and linear programming. IEEE Transactions on Information Theory, 51(11):3697–3717, 2005.
> > > > >
> > > > > [2]Simon De Givry. toulbar2, an exact cost function network solver. In 24ème édition du congrès annuel de la Société Française de Recherche Opérationnelle et d’Aide à la Décision ROADEF 2023, 2023.
> > > > >
> > > > > [3] Wang, Yaomin, et al. "NeuroLifting: Neural Inference on Markov Random Fields at Scale." arXiv preprint arXiv:2411.18954 (2024).
> > > > >
> > > > > [4] B. Andres, T. Beier, and J. H. Kappes. OpenGM2.
> > > > > http://hci.iwr.uni-heidelberg.de/opengm2/.

---

### Official Review · Reviewer_DrfU · 2025-11-01

**Soundness:** 2
**Presentation:** 3
**Contribution:** 2
**Rating:** 4
**Confidence:** 4

**Summary:**

**Summary**

This paper proposes a first-order method-based primal heuristic for the binary optimization problem. The main idea is to adopt a continuous extension of the optimization objective as well as a minimax formulation of the binary constraint. This reformulation is then solved with first-order methods for minimax optimization. Some numerical experiments demonstrate the efficiency of the proposed approach.

**Strengths:**

**Strength**

The paper has a clear motivation and is easy to follow. The idea of using a first-order method as a primal heuristic seems interesting.

**Weaknesses:**

**Weaknesses**

Although the idea seems interesting, I have some concerns regarding the algorithm design and the experiment setup.

1. Heuristic nature of the approach

   Although the paper establishes some theoretical results regarding the problem reformulation and the convergence of the gradient descent-ascent (GDA) method, there is actually nothing to say about the quality of the generated solution. In particular, the convergence analysis algorithm (PDBO) largely deviates from traditional GDA, and it only guarantees convergence to a binary solution. In other words, the proposed mechanism is essentially a primal heuristic that can be implemented by a first-order method. Therefore, the convergence analysis in the paper can be misleading.

2. Weak experiment evaluation

   In the experiments, the paper compares the proposed approach to several existing approaches from the literature. However, I find the following points unsatisfactory:

   - From the experiment, the proposed approach needs to be used jointly with other existing methods (e.g., ABS2) to give empirical speedup.
   - The method claims to be able to exploit GPU architecture. However, there is no time comparison between the CPU/GPU versions of the PDBO algorithm.
   - The selected optimization problems can also be solved with mathematical optimization solvers. It would be nice to see a comparison with other optimization solvers' performance.

Besides, the idea of using first-order methods to design MIP heuristics is not entirely new [1]. Solvers for mixed integer programming often have a solution polishing step that tries to sparsify a fractional solution using a sparsity-inducing loss function.

Overall, I find the proposed approach interesting, and I'm willing to raise my score if the authors can

- clarify the heuristic nature of the approach
- add experiments that demonstrate the scalability of PDBO on GPU
- add experiments that compare PDBO with an optimization solver (commercial or open-source).

**Questions:**

**Questions**

1. The formulation in equation (1) is quite general and seems compatible with the indicator function of constraints. Can the approach deal with linear constraints on $x$?
2. The algorithm seems to promote a binary solution only when the dual multipliers $y$ are negative. However, as shown in line 231, the initialization requires $y^0$ to be strictly positive. To me, this initialization seems to be trying to encourage more exploration of the feasible region. Could you elaborate more on this?
3. Currently, PDBO needs a projection that pushes the solution away from the stationary value $0.5$. What if you instead try escaping the stationary points by adding random noise and rounding the solution at the end?
4. Can your approach provide a dual certificate on the quality of the solution?

**Minor issues**

1. Line 75

   Problem (1) and problem (1) are sometimes inconsistent.

2. Line 110

   It may be better to add an introductory paragraph to the beginning of **Section 2**

3. Line 275

   What does it mean by linear time?

**References**

[1] Cacciola, M., Forel, A., Frangioni, A., & Lodi, A. (2025, June). The Differentiable Feasibility Pump. In *International Conference on Integer Programming and Combinatorial Optimization* (pp. 157-171). Cham: Springer Nature Switzerland.

**Details Of Ethics Concerns:**

N/A.

---

> ### Author Response · Authors · 2025-11-21
> **Rebuttal Part I**
>
> Thank you for the positive feedback on our work. We address the specific concerns below.
>
> > ### **W1. Heuristic nature of the approach**
> Although the paper establishes ...
>
> We thank the reviewer for this important discussion regarding the nature of our approach. We would like to clarify three key aspects in response:
>
> - **Practical Focus**: We focus on unconstrained binary optimization problems, which are NP-hard in general. Our primary goal is to develop a **practical and scalable** method for finding high-quality solutions efficiently. PDBO is designed as a powerful, **theory-inspired heuristic** that excels in scalability and benefits from GPU acceleration, making it particularly suitable for large-scale problems.
> - **Theoretical Contribution**: While we do not claim theoretical guarantees on solution quality (which is standard for NP-hard problems), our analysis provides a **non-trivial guarantee** that PDBO **converges** to feasible binary points. This is a significant practical advantage, as it **eliminates the need for ad-hoc rounding procedures** commonly required by prior works [2,3,4,5].
> - **Algorithmic Design**: PDBO builds upon the simple simultaneous GDA framework. The introduced modifications (Steps 3-4) are minimal, theoretically motivated, and crucial for preventing convergence to fractional stationary points while maintaining the method's first-order efficiency.
>
> Empirically, this design enables PDBO to achieve competitive solution quality with substantial speed advantages, especially on large-scale instances.
>
>
> > ### **W2. Weak experiment evaluation**
> > **W2.1.** From the experiment,...
>
>
> We thank the reviewer for this comment and would like to clarify the empirical results:
>
> - Across our four benchmark domains (Max-Cut, MIS, Max-k-SAT, and Max-k-Cut), **PDBO alone achieves superior solution quality and computational efficiency on MIS, Max-k-SAT, and Max-k-Cut compared to all other state-of-the-art baselines**. Its performance is **highly competitive on Max-Cut**, though slightly outperformed by the **specialized solver** ABS2.
> - This result is consistent with the design philosophies of the two methods. **ABS2** is a highly-specialized algorithm for Max-Cut, integrating **multiple sophisticated local search and genetic operations** that **iteratively refine the incumbent**. PDBO, in contrast, is a general-purpose, first-order method whose strength lies in its **scalability and rapid progress**, yet it still **generates high-quality solutions**, as evidenced by its performance trajectory in Figure 2.
> - For scenarios **demanding peak performance**, users can employ PDBO in conjunction with ABS2. The **hybrid PDBO+ABS2** strategy demonstrates a powerful synergy: PDBO's swift, high-quality initialization effectively bootstraps ABS2's intensive local search, leading to significant empirical speedups. This demonstrates that PDBO can effectively complement and accelerate even state-of-the-art specialized solvers.
>
>
>
>
> > **W2.2.**  The method claims to be...
>
> We thank the reviewer for highlighting the importance of empirically validating our method's GPU efficiency.
> - In response, we have conducted additional experiments comparing the **wall-clock time (in seconds)** of PDBO on both CPU and GPU over **1,000 GDA iterations**. The results in the following table demonstrate a substantial performance improvement. The GPU version achieves an **average speedup of ~15x**, robustly confirming its computational advantage. This aligns with the known strength of GPUs in accelerating the large-scale matrix operations that form the computational core of first-order methods [6].
>
> |Max-Cut|G67 |G70 |G72 |G77|G81|
> |-|-|-|-|-|-|
> |PDBO（GPU） |1.16 |0.93 |1.15 |1.47 |1.85 |
> |PDBO（CPU）|18.99 | 13.31|18.96 |24.24 |33.95 |
>
> To elaborate this, we have included a plot of objective value versus runtime in **Appendix D.2** of our revised manuscript, which clearly demonstrates that the GPU version is far more efficient.
>
>
>
> > **W2.3.** The selected optimization problems...
>
> Following the reviewer's suggestion, we have conducted additional experiments with Gurobi 10.0.2. The results demonstrate that PDBO achieves significantly better performance on Max-3-Cut, MIS, and MaxSAT instances. For the Max-Cut benchmarks, PDBO finds superior solutions for instances G70 and G81, while Gurobi performs better on G67, G72, and G77, albeit at a **substantially higher computational cost**. It is worth noting, **as highlighted in [7]**, that **recent versions of Gurobi have seen major improvements on QUBO and Max-Cut problems**, likely due to the integration of **specialized heuristics**. These new comparisons with Gurobi have been added to the revised manuscript. Moreover, we depict Gurobi's objective values against runtime in Figure 2, from which we conclude that PDBO achieves substantially better solutions than Gurobi for the majority of the runtime.

---

> > ### Author Response · Authors · 2025-11-21
> > **Rebuttal Part II**
> >
> > ### **Experiments on Gurobi**
> > | Max-Cut | G67-obj $\uparrow$|G67-time| G70-obj$\uparrow$|G70-time | G72-obj$\uparrow$|G72-time|G77-obj$\uparrow$|G77-time|G81-obj$\uparrow$|G81-time|
> > |-|-|-|-|-|-|-|-|-|-|-|
> > PDBO| 6872 | 2.5 | **9537** | 1.9 | 6906 | 2.0 | 9812 | 2.1 | **13852** | 3.3
> > Gurobi|**6944** | 151.9 | 9514 | 133.3 | **6990** | 171.6 | **9882** | 175.1 | 13848 | 179.8
> >
> > | Max-3-Cut | G67-obj $\uparrow$|G67-time| G70-obj$\uparrow$|G70-time | G72-obj$\uparrow$|G72-time|G77-obj$\uparrow$|G77-time|G81-obj$\uparrow$|G81-time|
> > |-|-|-|-|-|-|-|-|-|-|-|
> > PDBO | **8015** | 6.0 | **9999** | 3.3 | **8111** | 4.7 | **11467** | 5.0 | **16191** | 7.9 |
> > Gurobi| 7855 | 178.1 | **9999** | 0.7 | 8045 | 174.6 | 11101 | 179.9 | 15146 | 179.9 |
> >
> >
> >
> > | MIS | ($10^4$, 3)-obj$\uparrow$| ($10^4$, 3)-time |($10^4$, 100)-obj|($10^4$, 100)-time|($5\times10^4$, 3)-obj$\uparrow$|($5\times10^4$, 3)-time|($5\times 10^4$, 100)-obj$\uparrow$|($5\times 10^4$, 100)-time|
> > |-|-|-|-|-|-|-|-|-|
> > PDBO|$\bf{4431.9_{\pm  3.1 }}$ |$0.6 _{\pm  0.0 }$ |$\bf{603.0 _{\pm  3.3 }}$ |$3.6 _{\pm  0.3 }$ |$\bf{22127.3 _{\pm  10.3 }}$ |$1.1 _{\pm  0.1 }$ |$\bf{3006.2 _{\pm  6.2 }}$ |$14.6 _{\pm  0.6 }$ |
> > Gurobi |$4141.8 _{\pm  19.4 }$ |$147.2 _{\pm 24.8 }$ |$448.0 _{\pm  5.1 }$ |$0.1 _{\pm  0.0 }$ |$20144.3 _{\pm  181.7 }$ |$96.6 _{\pm  49.9 }$ |$2235.6 _{\pm  12.7 }$ |$0.1 _{\pm  0.0 }$ |
> >
> >
> > | MaxSAT |3CNF-obj$\downarrow$|3CNF-time| 4CNF-obj$\downarrow$|4CNF-time | 5CNF-obj$\downarrow$|5CNF-time|
> > |-|-|-|-|-|-|-|
> > PDBO |$\bf{1583.0 _{\pm  15.7 }}$ |$3.5 _{\pm  1.6 }$ |$\bf{1230.2 _{\pm  14.4 }}$ |$4.3 _{\pm  1.8 }$ |$\bf{1148.1 _{\pm  11.8 }}$ |$9.3 _{\pm  3.1 }$ |
> > Gurobi |$9322.6 _{\pm  64.2 }$ |$0.0 _{\pm  0.1 }$ |$9329.2 _{\pm  69.7 }$ |$0.0 _{\pm  0.0 }$ |$9310.7 _{\pm  75.4 }$ |$0.1 _{\pm  0.0 }$ |
> >
> >
> >
> > > **W2.4.** Besides, the idea of using first-order methods to design MIP heuristics is not entirely new [1]. Solvers for mixed integer programming often have a solution polishing step that tries to sparsify a fractional solution using a sparsity-inducing loss function.
> >
> > We appreciate the reviewer connecting our work to this relevant line of research. We agree that first-order methods are a powerful emerging tool for discrete optimization and would like to clarify the distinct contributions of PDBO:
> >
> > - **Scope & Goal**: Methods like the Differentiable Feasibility Pump [1] are designed for finding feasible solutions to **linearly constrained** MIPs. In contrast, PDBO is designed for a different objective: optimizing solution quality in **unconstrained** binary problems, where the primary challenge is navigating a complex, high-dimensional landscape.
> > - **Theoretical Framework**: PDBO is built upon a **principled primal-dual** framework with established max-min properties, providing a **theory-guided** approach to binary optimization.
> > - **Computational Efficiency**: PDBO is **entirely** based on first-order operations, making it **highly scalable on GPUs**. This contrasts with approaches like the differentiable feasibility pump, which often require solving linear programs as a subroutine, limiting their parallelization potential and scalability.
> >
> > > ### **W3.** I'm willing to raise my score if the authors can
> > > **W3.1**: clarify the heuristic nature of the approach
> >
> > Please refer to our reply to **W1**.
> > > **W3.2**: add experiments that demonstrate the scalability of PDBO on GPU
> >
> > Please refer to our reply to **W2.2**.
> > > **W3.3**:add experiments that compare PDBO with an optimization solver (commercial or open-source).
> >
> > Please refer to our reply to **W2.3**.
> >
> >
> >
> > ### Questions:
> >
> > > **Q1.** The formulation in equation (1) is quite general and seems compatible with the indicator function of constraints. Can the approach deal with linear constraints on $x$?
> >
> > Thanks for raising such an interesting point. You are correct that the indicator function for linear constraints, $\underset{x\in\{0,1\}^n}{\min}\mathbb{1}_{\small Ax \leq b},$ fits the form of (1). However, a significant challenge is that a **compact closed-form expression for the multilinear extension** of this indicator function is generally **not available**. This makes it difficult to apply our method directly to such constrained problems in its current formulation.

---

> ### Author Response · Authors · 2025-11-21
> **Rebuttal Part III**
>
> > **Q2.** The algorithm ...?
>
> We thank the reviewer for this insightful observation, which precisely captures the core design principle of PDBO. The initialization of the dual variable $y^0$ to a strictly positive value is indeed a **deliberate choice** to manage the optimization landscape.
> - The Lagrangian function $L(x,y):=f(x)+\sum\limits_{i=1}^ny_ig(x_i)$ has a curvature with respect to $x$ that is influenced by $y$. When $y$ is large and positive, the term $\sum\limits_{i=1}^ny_ig(x_i)$ dominates and, given the specific properties of our constraint function $g(x)$, **makes $L(x,y)$ convex with respect to $x$**. This initial convex phase facilitates **efficient exploration of the solution space** and helps avoid poor local minima early in the optimization.
> - As the GDA process proceeds, $y$ is updated and gradually decreases. When $y$ becomes non-positive, the landscape of $x$ becomes non-convex, which actively promotes the convergence of $x$ towards a binary solution. This design creates a smooth, automated transition from a **global exploration** phase to a final **binary exploitation** phase.
>
> We now clarify this in the revised manuscript.
>
> > **Q3.** Currently, ...?
>
> Thanks for this interesting suggestion.
> - Our current use of a deterministic projection is primarily a **theoretical device** to formally guarantee convergence to binary points. In practice, the stochasticity from random initialization and gradient estimation often **naturally prevents stabilization** at undesirable fractional points, which aligns with our empirical observations.
> - The suggested strategy of incorporating **random noise** and applying a final rounding is indeed a **valid and highly practical** alternative. We view this as a promising variant of PDBO and a valuable direction for future work.
>
> > **Q4.** Can your approach....?
>
> Thanks for raising this point. As we dicussed in our reply to Q2, when the initial value $y^0$ is large (e.g. $y^0 \geq -\lambda_{min}(W)$ for Max-Cut), the Lagrangian function $L(x,y)$ is **convex** in $x$. In this regime, we are able to **globally minimize $L(x,y)$** with respect to $x$ via classic gradient descent (albeit PDBO employ a single gradient descent step instead), the corresponding optimal value would provide a **valid dual bound** on the optimal objective value, serving as a dual certificate. We present the corresponding dual certificates of the Max-Cut problem in the following table.
>
> |Max-Cut|G67|G70|G72|G77|G81|
> |-|-|-|-|-|-|
> |primal bound | 6872| 9537 | 6906 | 9812| 13852 |
> |dual bound |8843.7|13891.5|8906.1|12767.7|18172.2|
>
> We clarify this in our revised manuscript in **Appendix D.5**.
>
> ### Minor issues
>
> > I1 Line 75..
>
> Thanks for catching this inconsistency in our manuscript. We carefully reviewed the entire document and ensured that all references to equations and problems follow consistent formatting, starting with capital "Problem" where appropriate, in our revised version.
>
> > I2 Line 110..
>
> Thanks, we now provide a brief introduction at the beginning of Section 2 in our revised manuscript.
>
> > I3 Line 275. .
>
> We thank the reviewer for seeking clarification on this point and apologize for the imprecise terminology in our original manuscript.
> - The reviewer is correct to seek clarification. In optimization theory, "linear convergence" refers to when after $k$ iterations, the distance to the optimum is $O(\rho^k)$ for some $0<\rho<1$, requiring $O(\log(\frac1\epsilon))$ iterations. Our proven rate of $O(\frac1\epsilon)$ iterations to reach an $\epsilon$-binary point is, in fact, a sublinear convergence rate.  We apologize for the imprecise language.
>
> We will correct the manuscript to state that the algorithm converges to a binary feasible point at a **sublinear rate**.
>
> ### **References**
>
> [1] Cacciola, M., Forel, A., Frangioni, A., & Lodi, A. (2025, June). The Differentiable Feasibility Pump. In International Conference on Integer Programming and Combinatorial Optimization (pp. 157-171). Cham: Springer Nature Switzerland.
>
> [2] Qiu, Yeqing, et al. "ROS: A GNN-based Relax-Optimize-and-Sample Framework for Max-k-Cut Problems." Forty-second International Conference on Machine Learning.
>
> [3] Schuetz, Martin JA, J. Kyle Brubaker, and Helmut G. Katzgraber. "Combinatorial optimization with physics-inspired graph neural networks." Nature Machine Intelligence 4.4 (2022): 367-377.
>
> [4] Shen, Zi-Song, et al. "Free-energy machine for combinatorial optimization." Nature Computational Science (2025): 1-11.
>
> [5] Ichikawa, Yuma. "Controlling continuous relaxation for combinatorial optimization." Advances in Neural Information Processing Systems 37 (2024): 47189-47216.
>
> [6] Lu, Haihao, and Jinwen Yang. "An Overview of GPU-based First-Order Methods for Linear Programming and Extensions." arXiv preprint arXiv:2506.02174 (2025).
>
> [7] Daniel Rehfeldt, Thorsten Koch, and Yuji Shinano. Faster exact solution of sparse maxcut and qubo problems. Mathematical Programming Computation, 15(3):445–470, 2023.

---

> > ### Comment · Reviewer_DrfU · 2025-11-24
> >
> > Thank you for the response. Based on the response, I raise my score to 6.

---

### Official Review · Reviewer_6DcA · 2025-11-03

**Soundness:** 3
**Presentation:** 3
**Contribution:** 3
**Rating:** 6
**Confidence:** 1

**Summary:**

The paper proposed a new primal-dual method for the unconstrained binary optimization problem via some convex smoothing for the binary constraints. Additionally, this method is also compatible with GPU computing.  Then, the authors show the performance of the proposed method both theoretically and numerically.

**Strengths:**

1. The idea is very clear and intuitive.

2. The proposed method is also GPU compatible.

3. The performance of the method is illustrated via both theory and experiments.

**Weaknesses:**

1. It seems the literature review part does not thoroughly discuss papers about solving binary optimization via continuous relaxation. Also see Q1 for more details.

**Questions:**

1. I am not familiar with the binary optimization literature. Could the authors provide more literature review? In particular, I wonder whether this idea —i.e., transforming binary constraints into continuous constraints via g(x)=0 — appears in the literature of binary optimization, not limited to GNN-based methods. This idea seems very direct, intuitive, and useful.

2. Why a good $g$ should be symmetric? I feel the asymmetry could also express some prior beliefs of the value of $x_i$, especially if we have some warm start solution in hand.

3. In general, how to select lambda for constrained problems, and alpha, beta as well?

some minor things:
a. Is the binary constraint in (3) a typo?

---

> ### Author Response · Authors · 2025-11-21
> **Rebuttal Part I**
>
> >**Q1.** I am not familiar with the binary optimization literature. Could the authors provide more literature review? In particular, I wonder whether this idea —i.e., transforming binary constraints into continuous constraints via g(x)=0 — appears in the literature of binary optimization, not limited to GNN-based methods. This idea seems very direct, intuitive, and useful.
>
> Thanks for raising this point.
> - **Literature review.** The presented literature review is not limited to continuous relaxation, but also include exact approaches [1,2,3], CPU-based heuristics[4,5] & GPU-based ones[6,7], and the learning-based methods [8,9,10,11]. Furthermore, we append two recent work [12, 13] on binary optimziation to enrich our literature review in the revised manuscript.
> - **Transforming binary constraints.** We thank the reviewer for this excellent question and the opportunity to clarify our work's position in the literature. The idea of transforming binary constraints via a continuous function $g(x)=0$ is indeed a classic technique.
>     - However, prior works (e.g., [13]) primarily treat this from a **penalty method** perspective. They add $g(x)$ as a penalty term to the objective using a **fixed, manually-tuned coefficient. This requires a delicate balance between the original objective and the penalty, often **lacks a principled update rule** for the coefficient, and **hinders generalization** across problem instances.
>     - Our method, PDBO, introduces a key conceptual shift: we formulate this constraint within a **primal-dual** framework. Here, the dual variable $-y_i$ acts as an **adaptive, variable-specific** penalty coefficient for $-g(x_i)\geq 0$. Crucially, these coefficients are not fixed but are dynamically and principledly updated according to the GDA rule. This formalism is not merely a different interpretation; it is the foundation that enables us to prove convergence to a binary solution, a theoretical guarantee that distinguishes PDBO from heuristic penalty methods.
>
> We now clarify this in the revised manuscript in Section 3.2.1.
>
>
>
> > **Q2**. Why a good $g$ should be symmetric? I feel the asymmetry could also express some prior beliefs of the value of , especially if we have some warm start solution in hand.
>
> We thank the reviewer for this insightful question. The choice of a symmetric $g$ is **designed for the general case** where no prior information is available, as it ensures the algorithm remains unbiased towards either 0 or 1. We agree that an asymmetric $g$ could be more useful when incorporating prior beliefs or warm-start solutions, guiding the optimization towards a preferred solution. We view this as a **promising and valuable direction** for future work.
>
>
> > **Q3**. In general, how to select lambda for constrained problems, and alpha, beta as well?
>
> Thanks for raising this point.
> - **Selection of $\lambda$ (Constraint Penalty)**: PDBO is primarily designed for unconstrained binary optimization. For constrained problems such as MIS, we **follow established practice** in the literature [8,9] by incorporating constraints into the objective via a penalty term. The value of $\lambda$ is adopted directly from these prior works to ensure a fair and consistent comparison. We emphasize that this penalization is a standard **pre-processing** step and **not the scope of our PDBO**.
> - **Selection of $\alpha, \beta$ (Stepsizes)**: The stepsizes $\alpha$ and $\beta$ are standard hyperparameters in classic first-order methods. In practice, a suitable configuration can be efficiently found via a grid search on a validation set or small-scale problems, which is a common and feasible approach in machine learning.
>
> We now clarify this in the revised manuscript.
>
> > **Q4.** Some minor things: a. Is the binary constraint in (3) a typo?
>
> Thank you for pointing this out. This is **not a typo**. As **stated in our manuscript**, "$g: [0,1] \rightarrow \mathbb{R}$ is a constraint function that enforces binarity of each variable $x_i \in [0,1]$". For example, when $g(x_i) := x_i(1-x_i)$, the condition $g(x_i) = 0$ is equivalent to $x_i \in \{0, 1\}$. We will clarify this in the revised manuscript.

---

> > ### Author Response · Authors · 2025-11-21
> > **Rebuttal Part II**
> >
> > ### **References**
> >
> > [1] Francisco Barahona, Michael Junger, and Gerhard Reinelt. Experiments in quadratic 0–1 programming. Mathematical programming, 44(1):127–137, 1989.
> >
> > [2] Svatopluk Poljak and Franz Rendl. Solving the max-cut problem using eigenvalues. Discrete Applied Mathematics, 62(1-3):249–278, 1995.
> >
> > [3] Daniel Rehfeldt, Thorsten Koch, and Yuji Shinano. Faster exact solution of sparse maxcut and qubo problems. Mathematical Programming Computation, 15(3):445–470, 2023.
> >
> > [4] Gintaras Palubeckis. Iterated tabu search for the unconstrained binary quadratic optimization problem. Informatica, 17(2):279–296, 2006.
> >
> > [5] Yang Wang, Zhipeng Lu, Fred Glover, and Jin-Kao Hao. Probabilistic grasp-tabu search algorithms for the ubqp problem. Computers & Operations Research, 40(12):3100–3107, 2013.
> >
> > [6] Koji Nakano, Daisuke Takafuji, Yasuaki Ito, Takashi Yazane, Junko Yano, Shiro Ozaki, Ryota Katsuki, and Rie Mori. Diverse adaptive bulk search: a framework for solving qubo problems on multiple gpus. In 2023 IEEE International Parallel and Distributed Processing Symposium Workshops (IPDPSW), pp. 314–325. IEEE, 2023.
> >
> > [7] Zi-Song Shen, Feng Pan, Yao Wang, Yi-Ding Men, Wen-Biao Xu, Man-Hong Yung, and Pan Zhang. Free-energy machine for combinatorial optimization. Nature Computational Science, pp. 1–11,2025.
> >
> > [8] Schuetz, Martin JA, J. Kyle Brubaker, and Helmut G. Katzgraber. "Combinatorial optimization with physics-inspired graph neural networks." Nature Machine Intelligence 4.4 (2022): 367-377.
> >
> > [9] Ichikawa, Yuma. "Controlling continuous relaxation for combinatorial optimization." Advances in Neural Information Processing Systems 37 (2024): 47189-47216.
> >
> > [10] Qiu, Yeqing, et al. "ROS: A GNN-based Relax-Optimize-and-Sample Framework for Max-k-Cut Problems." Forty-second International Conference on Machine Learning.
> >
> > [11] Jan T¨onshoff, Berke Kisin, Jakob Lindner, and Martin Grohe. One model, any csp: Graph neural networks as fast global search heuristics for constraint satisfaction. In Proceedings of the Thirty-Second International Joint Conference on Artificial Intelligence, IJCAI-23
> >
> > [12] Chen, C., Chen, R., Li, T. et al. A Monte Carlo Policy Gradient Method with Local Search for Binary Optimization. Math. Program. (2025).
> >
> > [13] Zhou, Shenglong, et al. "Sharp-Peak Functions for Exactly Penalizing Binary Integer Programming." arXiv preprint arXiv:2509.00895 (2025).

---

### Official Review · Reviewer_TYvk · 2025-11-03

**Soundness:** 3
**Presentation:** 3
**Contribution:** 2
**Rating:** 4
**Confidence:** 2

**Summary:**

In this paper, the authors propose a primal-dual approach to solve quadratic unconstrained binary optimization (QUBO) problems. The idea is to consider a continuous relaxation of the original unconstrained version and introduce additional constraints to enforce integrality. Then a gradient descent-ascent algorithm is implemented to solve the primal-dual reformulation of the constrained version. Empirical evaluations on 4 classical QUBO problems show the proposed algorithm (PDBO) outperforming existing baselines.

**Strengths:**

1. The paper is well-written and easy to read. The primal-dual reformulation makes intuitive sense.
2. On large graph instances, PDBO finds better feasible solutions within a short time compared to baseline methods.

**Weaknesses:**

1. The empirical evaluations for Max-Cut and Max-k-Cut only presents 5 large instances in the main paper. The remaining instances present a less clear picture, especially for instances under 10000 nodes compared to the ABS2 baseline.

2. As the method is based on continuous relaxation, the comparison does not have any existing relaxation-based baselines. Another important missing baseline is exact solvers such as Gurobi. In terms of learning-based baselines, some recent works are missing, e.g., [1]. Adding those baselines would strengthen the submission.

[1]: Chen, Ming, et al. "Learning to solve quadratic unconstrained binary optimization in a classification way." Advances in Neural Information Processing Systems 37 (2024): 114478-114509.

**Questions:**

1. What are the reasonings behind the three conditions for the function $g$ on lines 173-175?
2. In the algorithm 1, line 3 has one condition for the partial derivative. Why is this condition necessary?
3. In Table 1, PDBO+ABS2 hybrid strategy exhibits vastly different runtime. What makes G67 and G72 take so much longer than the other three instances?

---

> ### Author Response · Authors · 2025-11-21
> **Rebuttal Part I**
>
> > **W1**. The empirical evaluations for Max-Cut and Max-k-Cut only presents 5 large instances in the main paper. The remaining instances present a less clear picture, especially for instances under 10000 nodes compared to the ABS2 baseline.
>
> Thank you for this observation.
>
> - We acknowledge that ABS2 demonstrates strong performance on smaller Max-Cut instances. It is important to note that ABS2 is a highly **specialized solver for Max-Cut**. It combines **multiple local search** mechanisms—such as Greedy, MaxMin, and CyclicMin search—with genetic operations to **iteratively refine solutions**. However, we note that ABS2 is not applicable to the more general Max-*k*-Cut problem. In contrast, PDBO is designed to be more **general and scalable**, capable of addressing a broader class of combinatorial optimization problems. Its efficiency on large-scale instances stems from a **single-step, first-order** optimization strategy that offers significant scalability.
> - Regarding the smaller instances (with at most 10,000 nodes), PDBO alone matches or outperforms ABS2 on 16 of the 66 remaining benchmarks. More notably, the hybrid PDBO+ABS2 method—which leverages PDBO's rapid initialization followed by ABS2's intensive local search—matches or surpasses the performance of standalone ABS2 on 44 out of these 66 instances.
>
> We clarify this in **Appendix D.1** of our revised manuscript.
>
> > **W2**. As the method is based on continuous relaxation, the comparison does not have any existing relaxation-based baselines. Another important missing baseline is exact solvers such as Gurobi. In terms of learning-based baselines, some recent works are missing, e.g., [1]. Adding those baselines would strengthen the submission.
>
> We thank the reviewer for these insightful suggestions regarding additional baselines.
>
> - **Regarding relaxation-based baselines**: We would like to clarify that several of our **existing baselines** are, in fact, **founded on continuous relaxation** strategies. For instance, PIGNN relaxes the binary constraints of a QUBO problem $\min x^\top Hx$ by allowing $x\in\{0,1\}^n$ to $x\in[0,1]^n$. Similarly, methods like CRA, FEM, and ROS are explicitly designed around continuous relaxations, with ROS specifically employing a multilinear extension for the Max-k-Cut problem.
>
> - **Regarding the exact solver (Gurobi)**: Following the reviewer's suggestion, we have conducted additional experiments with Gurobi 10.0.2. The results demonstrate that PDBO achieves significantly better performance on Max-3-Cut, MIS, and MaxSAT instances. For the Max-Cut benchmarks, PDBO finds superior solutions for instances G70 and G81, while Gurobi performs better on G67, G72, and G77, albeit at a **substantially higher computational cost**. It is worth noting, **as highlighted in [2]**, that **recent versions of Gurobi have seen major improvements on QUBO and Max-Cut problems**, likely due to the integration of **specialized heuristics**. This context underscores a key strength of PDBO: it is a general-purpose approach that remains competitive across a diverse set of unconstrained binary optimization problems without relying on problem-specific customizations. These new comparisons with Gurobi have been added to the revised manuscript. Moreover, we depict Gurobi's objective values against runtime in **Figure 2**, from which we conclude that PDBO **achieves substantially better solutions than Gurobi for the majority of the runtime.**
>
> - **Regarding the learning-based baseline [1]**: We appreciate the reviewer for bringing this relevant work to our attention.
>   - However, the official code for [1] is not publicly available (the provided GitHub repository is empty), which makes a direct and fair empirical comparison infeasible within the rebuttal period.
>   - To ensure a robust comparison, our study already includes several recent, state-of-the-art learning-based methods, such as **ROS (*ICML* 2025), CRA (*NeurIPS* 2024), ANYCSP (*IJCAI* 2023), and PIGNN (*Nature Machine Intelligence* 2022)**. We believe this selection constitutes a strong and diverse set of baselines that adequately represents the current landscape of learning-based solvers.

---

> ### Author Response · Authors · 2025-11-21
> **Rebuttal Part II**
>
> ### **Experiments on Gurobi**
> | Max-Cut | G67-obj $\uparrow$|G67-time| G70-obj$\uparrow$|G70-time | G72-obj$\uparrow$|G72-time|G77-obj$\uparrow$|G77-time|G81-obj$\uparrow$|G81-time|
> |-|-|-|-|-|-|-|-|-|-|-|
> PDBO| 6872 | 2.5 | **9537** | 1.9 | 6906 | 2.0 | 9812 | 2.1 | **13852** | 3.3
> Gurobi|**6944** | 151.9 | 9514 | 133.3 | **6990** | 171.6 | **9882** | 175.1 | 13848 | 179.8
>
> | Max-3-Cut | G67-obj $\uparrow$|G67-time| G70-obj$\uparrow$|G70-time | G72-obj$\uparrow$|G72-time|G77-obj$\uparrow$|G77-time|G81-obj$\uparrow$|G81-time|
> |-|-|-|-|-|-|-|-|-|-|-|
> PDBO | **8015** | 6.0 | **9999** | 3.3 | **8111** | 4.7 | **11467** | 5.0 | **16191** | 7.9 |
> Gurobi| 7855 | 178.1 | **9999** | 0.7 | 8045 | 174.6 | 11101 | 179.9 | 15146 | 179.9 |
>
>
>
> | MIS | ($10^4$, 3)-obj$\uparrow$| ($10^4$, 3)-time |($10^4$, 100)-obj|($10^4$, 100)-time|($5\times10^4$, 3)-obj$\uparrow$|($5\times10^4$, 3)-time|($5\times 10^4$, 100)-obj$\uparrow$|($5\times 10^4$, 100)-time|
> |-|-|-|-|-|-|-|-|-|
> PDBO|$\bf{4431.9_{\pm  3.1 }}$ |$0.6 _{\pm  0.0 }$ |$\bf{603.0 _{\pm  3.3 }}$ |$3.6 _{\pm  0.3 }$ |$\bf{22127.3 _{\pm  10.3 }}$ |$1.1 _{\pm  0.1 }$ |$\bf{3006.2 _{\pm  6.2 }}$ |$14.6 _{\pm  0.6 }$ |
> Gurobi |$4141.8 _{\pm  19.4 }$ |$147.2 _{\pm 24.8 }$ |$448.0 _{\pm  5.1 }$ |$0.1 _{\pm  0.0 }$ |$20144.3 _{\pm  181.7 }$ |$96.6 _{\pm  49.9 }$ |$2235.6 _{\pm  12.7 }$ |$0.1 _{\pm  0.0 }$ |
>
>
> | MaxSAT |3CNF-obj$\downarrow$|3CNF-time| 4CNF-obj$\downarrow$|4CNF-time | 5CNF-obj$\downarrow$|5CNF-time|
> |-|-|-|-|-|-|-|
> PDBO |$\bf{1583.0 _{\pm  15.7 }}$ |$3.5 _{\pm  1.6 }$ |$\bf{1230.2 _{\pm  14.4 }}$ |$4.3 _{\pm  1.8 }$ |$\bf{1148.1 _{\pm  11.8 }}$ |$9.3 _{\pm  3.1 }$ |
> Gurobi |$9322.6 _{\pm  64.2 }$ |$0.0 _{\pm  0.1 }$ |$9329.2 _{\pm  69.7 }$ |$0.0 _{\pm  0.0 }$ |$9310.7 _{\pm  75.4 }$ |$0.1 _{\pm  0.0 }$ |
>
> ### **Questions**
> >Q1. What are the reasonings behind the three conditions for the function on lines 173-175?
>
> Thanks for raising this point.
> * The first condition is to ensure that $g(x_i)=0$ enforces $x_i$ to be binary, and the convexity with $g'(x_i)\neq 0$ guarantees that $g(x_i)<0$ for fractional $x$.
> * The second condition is to ensures differentiability such that the GDA can be applied.
> * The last condition ensures symmetric. We do not assume prior knowledge, and penalize fractionality symmetrically. Conclusively, $-g$ can be regarded as a metric (or penalty) for fractionality.
> * Moreover, the convexity is for practically improvement. By setting $y^0$ sufficiently, the Lagrangian function $L(x,y)$ is convex in $x$ and therefore encourages exploration in the beginning.
>
> We now clarify this in our revised manuscript.
>
> >Q2. In the algorithm 1, line 3 has one condition for the partial derivative. Why is this condition necessary?
>
> Thank you for this question. The condition on Line 3 of Algorithm 1 is designed to **perturb the solution** if it stagnates near $\frac{1}{2}$ in any coordinate. When the **partial derivative** with respect to a variable is near zero, the gradient-based update for that dimension becomes negligible, causing the optimization to stall. By applying a small perturbation in such cases, we help the algorithm escape these flat regions and continue progressing toward a binary solution. We now clarify this in our revised manuscript.
>
> >Q3. In Table 1, PDBO+ABS2 hybrid strategy exhibits vastly different runtime. What makes G67 and G72 take so much longer than the other three instances?
>
> We thank the reviewer for raising this question. As we mentioned in our manuscript, "**The columns labeled “Time” indicates the earliest time at which the method attains the solution corresponding to the reported objective value.**" Thus, the recorded "Time" in Table 1 is the earliest time the best solution is found within the 180-second budget. For G67 and G72, the hybrid method progressively improves the solution, hence the longer time. In the other instances, the initial solution from PDBO is of such high quality that it cannot be improved upon, resulting in a very short recorded time.
>
>
> ### **Reference**
> [1] Chen, Ming, et al. "Learning to solve quadratic unconstrained binary optimization in a classification way." *Advances in Neural Information Processing Systems* 37 (2024): 114478-114509.
>
> [2] Daniel Rehfeldt, Thorsten Koch, and Yuji Shinano. Faster exact solution of sparse maxcut and qubo problems. *Mathematical Programming Computation*, 15(3):445–470, 2023.

---

### Author Response · Authors · 2025-12-02
**Summary of Rebuttal and Revisions**

We thank the reviewers for their constructive feedback. Below, we summarize the primary concerns raised and our corresponding revisions.

> ### **Concern 1**. Comparison against Gurobi

**Reviewers**: TYvk, DrfU, s1Wx

**Response**: We conducted new experiments comparing PDBO against the exact solver Gurobi. The results, presented in Tables 1–4 and Figure 2, demonstrate PDBO's competitive effectiveness and superior computational efficiency, especially on large-scale instances.


> ### **Concern 2**. Evaluation on Maximum A Posteriori (MAP) problems

**Reviewer**: s1Wx

**Response**: Following this suggestion, we extended our evaluation to two MAP benchmarks:
  - The protein-prediction dataset (suggested by the reviewer).
  - The UAI 2022 Competition Grids dataset.
Results, detailed in Tables 8 & 9 (Appendix D.3), show that PDBO consistently outperforms established baselines, including the competition winner Toulbar2.


> ### **Concern 3**. Clarification on the distinction of PDBO

**Reviewers**: All

**Response**: We have clarified the methodological position of PDBO from two key angles:
- *PDBO vs. Continuous Relaxation Methods*: We highlight that PDBO dynamically adjusts the penalty coefficients, instead of a static one.
- *PDBO vs. Continuation Methods*: We highlight the theoretical advantage of PDBO’s larger extension space. We explain the principled, automatic dual updates in PDBO versus predefined penalty schedules.

These points are supported by both theoretical discussion and new practical comparisons in Appendices D.4 & D.5. All new experiments, tables, and clarifications have been integrated into the revised manuscript.

---

### Meta-Review · Area_Chair_BcQ6 · 2026-01-05

**Summary:**

The main result of the paper is a simple, generic approach to solving (non-convex) binary optimization problems: an exact primal-dual formulation (see Theorem 1) optimized using first-order methods. In the paper and rebuttal to 6DcA, they mention the main novelty of the method is the more principled exact formulation, compared to penalized relaxations of previous comparable approaches.

The reviewers appreciate the simplicity and generic applicability of the method. They also highlight concerns re: novelty and evaluation. I agree with the reviewers that for a heuristic paper such as this one, the main way to judge acceptance is a comprehensive evaluation. Specifically, for a heuristic that is closely-related to other approaches in the literature (penalized Lagrangian + first-order method is not a new idea), the "technical novelty" of exact primal-dual formulation in this paper should clearly improve on competing baselines. While the authors did discuss how continuous relaxation motivate some benchmarks in their response to DrfU, and provided a new comparison in the appendix, ultimately I am left unconvinced that the novelty is useful enough on its own to merit acceptance.

If the authors believe their perspective is really the right way to do primal-dual formulation + first-order method, I would suggest evaluating against competing such approaches in an ablation test, and show that their algorithm outperforms previous strategies.

**Reviewer Concerns:**

Some of the decisions made by the authors were defended, but ultimately I would have liked to see the authors engage more directly with the novelty and evaluation concerns, specifically against other baselines that use the continuous relaxation + first-order method strategy. If the novelty highlight is in the formulation in Section 3.1, then it should be more thoroughly compared to other similar approaches. While the authors take a step towards doing this in Section D.4, I do think the reviewers would want a more thorough evaluation: are the performance gains seen in Tables 1-4 due to the combination of (A) exact primal-dual formulation and (B) first-order method, or would other continuous relaxations + (B) also get similar performance boosts?

**Reviewer Scores:**

I think the reviewers who raised conceptual novelty and evaluation concerns would leave with similar thoughts as I have written (s1Wx expressed willingness to raise score, but it seemed tentative).

---

### Decision · Program_Chairs · 2026-01-26

Reject